# Self-supervised Color Generalization in Reinforcement Learning

**Matthias Weissenbacher**  *wbmatthias@gmail.com*
*Riken Center for Advanced Intelligence Project*
*Pyr-SAI Labs*
*Japan*

**Evangelos Routis**  *evangelos_routis@alumni.brown.edu*
*Causaly*
*London*
*United Kingdom*

**Yoshinobu Kawahara**
*Riken Center for Advanced Intelligence Project*
*Osaka University*
*Japan*

**Reviewed on OpenReview:** *https://openreview.net/forum?id=4OnOPLRI8H*

## Abstract

A challenge in reinforcement learning lies in effectively deploying trained policies to handle out-of-distribution data and environmental variations. Agents observing pixel-based image data are generally sensitive to background distractions and color changes. Commonly, color generalization is achieved through data augmentation. In contrast, we propose a color-invariant neural network layer that adopts distinct color symmetries in a self-supervised fashion. This allows for color sensitivity while achieving generalization. Our approach is based on dynamic-mode decomposition, which also accommodates spatial and temporal symmetries; we discuss the controlled breaking of the latter. We empirically evaluate our method in the Minigrid, Procgen, and DeepMind Control suites and find improved color sensitivity and generalisation.

## 1 Introduction

Reinforcement learning has seen tremendous success, but ensuring it works well on unfamiliar data is still a challenge. The original approach to generalisation in image-based RL, which used randomized image augmentations (Laskin et al., 2020; Yarats et al., 2021b), was remarkably successful despite its brute-force nature. However, using image-based data augmentation can lead to over-regularisation to specific augmentations and thus may perform poorly when applied to certain environments. The more evolved methodology of selecting specific image augmentations in a self-supervised way leads to increased performance and alleviates some aspects of over-regularisation (Raileanu et al., 2020; Hansen & Wang, 2021).

However, finding the balance between color sensitivity and color generalisation remains an open challenge, which is not addressed by the above-mentioned methods. Color sensitivity refers to the agent's ability to take action based on specific colors of objects in the environment, such as whether to stop or go at a red or green traffic light, see Figure (1). In contrast, it is imperative that the agent's decision-making process remains invariant to color changes in certain environmental factors, e.g., the color of houses, trees, or non-emergency vehicles in the context of autonomous vehicles.

An alternative to data augmentation is utilizing symmetries in RL to improve generalization (Tang & Ha, 2021; van der Pol et al., 2020; Weissenbacher et al., 2022). Specifically, reinforcement learning may benefit from

both local and global symmetries, which preserve a particular structure or property within a neighborhood of a point and throughout the entire space, respectively. In the above example, a house or other vehicles are generally local, that is, they occupy a small area of the entire visual field of the agent. Enforcing local symmetries through data augmentation becomes impractical as the sample size required grows exponentially with the number of image patches under consideration. This makes the approach highly inefficient and computationally expensive.

Our approach introduces the capability to develop a self-supervised understanding of these local and global color symmetries. Our **C**olor-**i**nvariant **L**ayer (CiL) is based on the Dynamic-mode-decomposition, which theoretically incorporates spatial and temporal permutation invariance (PI). We do not make use of spatial PI in our model and break temporal symmetries in the case of frame-stacks as inputs, so that the time-wise order of frames is relevant.

In this work, we introduce a novel framework of self-supervised selection of features. We apply the above-mentioned paradigm of symmetries to self-supervised selection of color features to replace image augmentation with random-conv and color-jitter filters. Consider the conventional training of an agent with randomly color-augmented images (see Figure (1). Depending on the task, one could select a subset of these images that are more suitable. Our method effectively performs this selection of 'useful' color augmentations in a self-supervised manner. CiL admits different inductive biases categories (I)-(III) admitting an increased amount of trainable parameters corresponding to specific subsets of augmented images. The most suitable bias is selected during training in a self-supervised fashion.

In section 3 we introduce both the architecture for the CiL layer as well as provide formal propositions which validate its symmetry properties. In section 4.1, we show that in the simple example of adding a safe blue river (water) to the mini-grid LavaCrossing environment data augmentation - random-conv and color-jitter - is detrimental to learning, while CiL is able to navigate the color-sensitive environment. Moreover, we find improved color generalisation compared to the baseline. In section 4.2 we perform our main experiments on the Procgen environment. The code is made public at GitHub.

## 2 Background

### 2.1 Reinforcement Learning & Symmetries

**Reinforcement Learning.** A Markov Decision Process (MDP) is a mathematical framework for modeling decision-making problems in stochastic environments. MDPs are characterized by a tuple $(\mathcal{S}, \mathcal{A}, \mathcal{P}, \mathcal{R}, \gamma)$, where $\mathcal{S}$ is a finite set of states, $\mathcal{A}$ is a finite set of actions, $\mathcal{P}$ is the transition probability function, $\mathcal{R}$ is the reward function, and $\gamma \in [0, 1)$ is the discount factor. In RL, one aims to learn optimal decision-making policies in MDPs. A policy, denoted as $\pi : \mathcal{S} \to \mathcal{A}$, is a mapping from states to actions. The optimal policy $\pi^*$ maximizes the expected cumulative discounted reward, given by the value function $V^\pi(s) = \mathbb{E}\left[\sum_{t=0}^{\infty} \gamma^t \mathcal{R}(s_t, a_t) \mid s_0 = s, a_t \sim \pi(\cdot \mid s_t)\right]$, where the expectation is taken over the sequence of states and actions encountered by following the policy $\pi$. The optimal policy $\pi^*$ is the one that satisfies $V^{\pi^*}(s) \geq V^\pi(s)$ for all $s \in \mathcal{S}$ and any other policy $\pi$.

**Symmetry groups**. Symmetry transformations are special cases of mathematical groups, i.e. sets that contain the identity element and are closed under multiplication and inverses. In this work we encounter the group of orthogonal matrices, in particular the three-dimensional rotation group $SO_3$, i.e. the special orthogonal group in dimension 3. Any element of the latter can be written in terms of three independent parameters i.e. the rotation angles along the x,y and z-axis, respectively, called the Euler angles.

**Invariance**. Invariance is a foundational concept in understanding how functions respond to symmetries of their inputs. Before defining these concepts, we introduce some notation. Let $f$ be a function that maps elements from space $\mathcal{S}$ to space $\mathcal{S}'$. Let $G$ be a symmetry group acting on $\mathcal{S}$ and let us denote by $g$ an individual transformation in $G$. We shall denote the action of the transformation $g$ on an element $s$ of $\mathcal{S}$ by $g \cdot s$. A function $f$ is invariant with respect to a set of transformations (symmetry group $G$) if the application of any transformation from this set to its input does not change the function's output. Mathematically, this is expressed as $f(g \cdot s) = f(s)$ for every transformation $g \in G$.

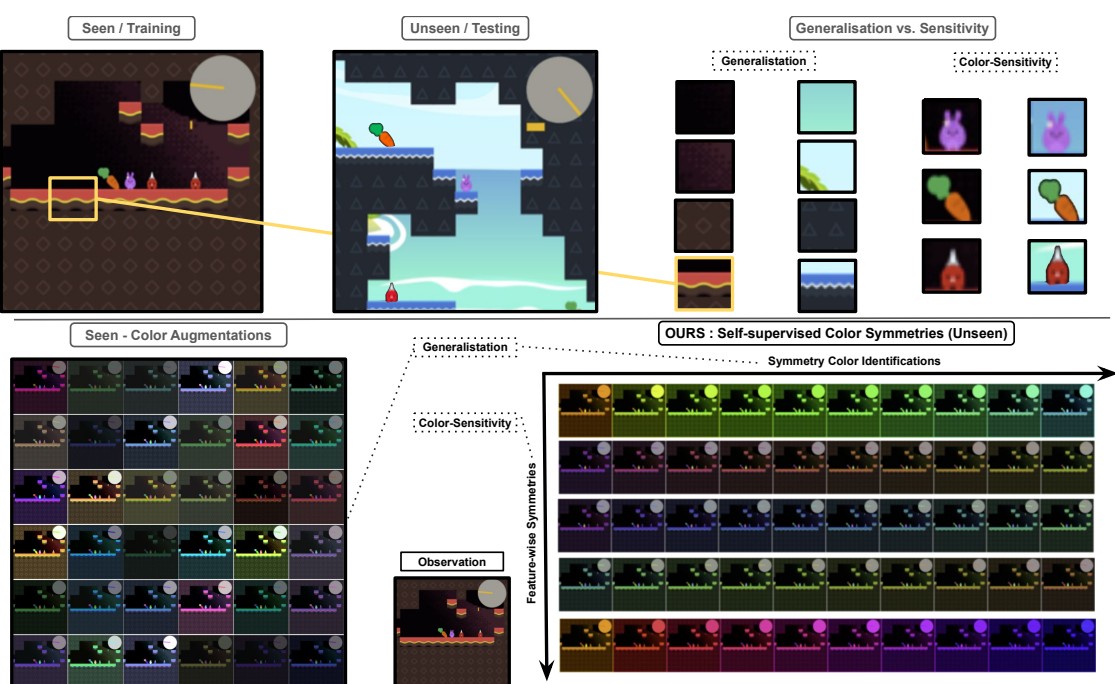

**Figure 1:** Top: Overview of the balance of color generalisation vs. color sensitivity required for RL tasks, in the example of the Jumper environment of the Procgen suite. Bottom: Conventional random color-augmentation i.e. color-jitter (left) compared to OUR self-supervised learning of color-symmetries (right), in the example of Category II color symmetries -horizontal axis- per each feature of the model - vertical axis-. The vertical axis shows different features-wise choices of Category II symmetries. The horizontal axis shows identical images under the particular Category II symmetry. In other words, CiL produces the same feature-wise output if the input image varies according to horizontal axis.

## 2.2 Dynamic Mode Representation

The dynamic-mode-decomposition (DMD) algorithm fits a linear operator $\mathcal{K}$ that advances the state of a system, $s \in \mathbb{R}^n$ forward in time (Schmid, 2010; Tu et al., 2013; Kutz et al., 2016; Rowley et al., 2009; Williams et al., 2015). Thus, the linear dynamical system is given by

$$s_{t_{i+1}} \approx \mathcal{K}\, s_{t_i} \;\;,\;\; \text{for}\;\; i = 1, \ldots, N \tag{1}$$

The operator $\mathcal{K}$ is an approximation of the Koopman operator restricted to observables given by direct measurements of the state $s$. In practice, the operator $\mathcal{K}$ is computed from a collection of snapshot pairs of the system $\{s_{t_{i+1}}, s_{t_i}\}_{i=1,\ldots,N}$. In principle, for DMD the times need not be sequential or evenly spaced, however this is the case for most RL settings. These snapshots are arranged into two data matrices, $S$ and $S'$ as

$$S = \left(s_{t_1}, s_{t_2} \ldots, s_{t_N}\right) \;\;,\;\; \text{and}\;\; S' = \left(s_{t_2}, s_{t_3} \ldots, s_{t_{N+1}}\right) \tag{2}$$

thus equation 1 may be rewritten as $S' \approx \mathcal{K}\,S$. The computation of operator $\mathcal{K}$ from given transition dynamics data can be reformulated as an optimization problem

$$\mathcal{K} = \arg\min_K |S' - K\,S|_F = S'\,S^\dagger \tag{3}$$

where $|\cdot|_F$ is the Frobenius norm and $\dagger$ denotes the pseudo-inverse. The pseudo-inverse may be computed by applying a singular value decomposition (SVD). In the case of image data with color features we stack them in the time-direction as

$$\begin{aligned} S &= \left(s_{t_1}, s_{t_1}, s_{t_1}, \ldots, s_{t_N}, s_{t_N}, s_{t_N}\right) \\ S' &= \left(s_{t_2}, s_{t_2}, s_{t_2}, \ldots, s_{t_{N+1}}, s_{t_{N+1}}, s_{t_{N+1}}\right) \end{aligned} \tag{4}$$

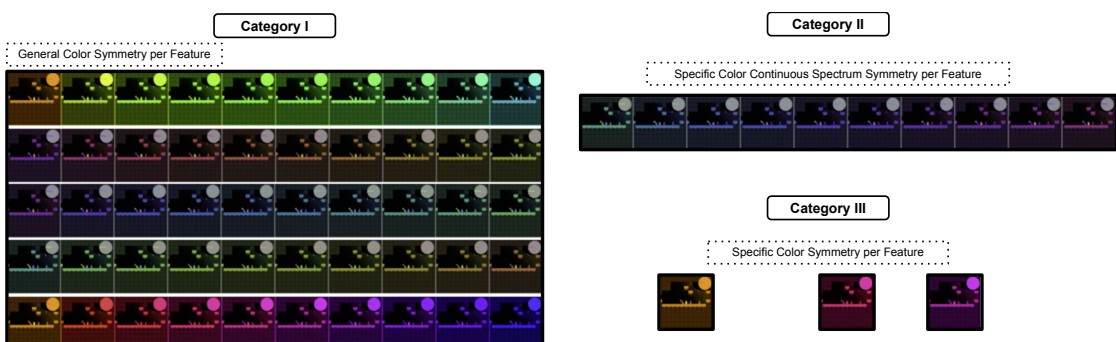

**Figure 2:** Three categories of the feature-wise color symmetries of CiL in the example of the Jumper environment of the Procgen Suite (Cobbe et al., 2020).

A time sequence of data is very commonly modified by an invertible linear transformation acting on each state as well as an invertible linear transformation acting on the time-series as

$$S \to \mathcal{R}S \ , \quad S \to S\mathcal{L} \tag{5}$$

$\forall i = 1, \ldots, N$ and where $\mathcal{R} \in \mathbb{R}^{n \times n}$ and $\mathcal{L} \in \mathbb{R}^{3N \times 3N}$. We restrict ourselves to orthogonal transformations i.e. $\mathcal{R}^{-1} = \mathcal{R}^T$ and $\mathcal{L}^{-1} = \mathcal{L}^T$.

### 2.2.1 Leading Rank Representation

Since the matrix operator $\mathcal{K}$ typically admits $n^2$ elements, for high-dimensional data, it is intractable to represent this operator. Instead, one may compute its leading spectral decomposition. In particular, for time series pixel data, the sequence length $N$ is much shorter (in the context of RL) than the number of pixels in the 2D image, thus the matrices $S$ and $S'$ have far more rows than columns. It follows that the size of the set of eigenvectors of $\mathcal{K}$ corresponding to its nonzero eigenvalues is at most $N$ times the number of colors (input channels of each image). In practice, the effective rank $r$ of the data matrices $S$ and $S'$ and hence the operator $\mathcal{K}$, generically is even lower.

The algorithm of the leading order rank approximation (rDMD) can be outlined as follows. Instead of computing $\mathcal{K}$ in equation 3, we may project the latter onto its first $r$ singular vectors. With $U_r, \Sigma_r, V_r$ the rank-$r$ restricted singular value decomposition of $S$ one can approximate the pseudo-inverse as $S \approx U_r \Sigma_r V_r^*$. Then the operator $\mathcal{K}_r$ can be defined as

$$\text{rDMD:} \quad \mathcal{K}_r := U_r^* \mathcal{K} U_r = U_r^* \, S' \, V_r \, \Sigma_r^{-1} \ . \tag{6}$$

The leading spectral decomposition of $\mathcal{K}$ may be approximated from the spectral decomposition of the much smaller $\mathcal{K}_r$ as $\mathcal{K}_r W = W \Lambda$ [1]. The diagonal matrix $\Lambda$ contains the DMD eigenvalues, which correspond to eigenvalues of the high-dimensional matrix $\mathcal{K}$.

## 3 An Adaptive Color Invariant Layer

In this section, we delve into the architecture of our color-invariant layer (CiL), offering a detailed technical overview. Subsequently, in section 3.1, we provide its theoretical underpinning. Central to CiL is the strategy of partitioning the image into patches, drawing parallels with vision transformers (Dosovitskiy et al., 2020). For each of these patches, we then compute the Singular Value Decomposition (SVD), or in some cases, the mean over adjacent patches, as depicted in Figure (3). This approach is twofold: firstly, it minimizes the computational demands of SVD; secondly, it paves the way for achieving local color invariance.

Conceptually, we divide color symmetries, both local and global, into three distinct mathematical categories (I)-(III). Very general invariance (I); (II) the model can in a self-supervised way learn a specific color spectrum

---

[1]The eigenvectors of $\mathcal{K}_r$ are not the same as the eigenvectors of $\mathcal{K}$- there is a precise relation between the two (Brunton, 2019)

- per each 3-channel[2] - for which it is invariant, see Figure (1) and Figure (2); for type (III) the model in a self-supervised fashion learns a pair of color-transformation which it understands to be the same, i.e is invariant under- per each 3-channel -. Category (I) is comparable to random-conv and color-jitter filters in that it is not color-sensitive. Categories (II)-(III) increasingly trade off generalisation to gain more color sensitivity.

For simplicity, let us discuss the single frame case first i.e. $N = 1$ for which the core functionality of the CiL layer is given by

$$\boxed{\text{CiL} \; : \quad S \quad \mapsto \quad S \; \mathcal{W} \; V} \tag{7}$$

where V is computed by the SVD of S, and $\mathcal{W}$ is a 3x3 matrix of trainable weights.[3] The choice of the latter results in the following categories:

- **Category I :** $\mathcal{W}$ is the identity matrix i.e without any trainable parameters, equation 7 is invariant under general orthogonal transformations.

- **Category II :** For $\mathcal{W} \in SO_3$, i.e. in the general rotations group equation 7 is invariant under a continuous one-parameter family of color transformations. For a given element in $SO_3$ corresponds to a rotation in three-dimensional color space and the layer is symmetric w.r.t. any other oration along the same axis. The specific axis is chosen by setting the three Euler angles i.e. independent parameters in $\mathcal{W}$.

- **Category III :** For $\mathcal{W}$ a symmetric (real) $3 \times 3$ matrix , equation 7 is invariant under a set of 4 (or 8) specific color transformations which are chosen by setting the parameters in $\mathcal{W}$.

For a visualisation see Figure 2. For multiple frames i.e. $N > 1$ we need to take into account the time-series nature according to equation 4.

$$\boxed{\text{CiL-Stack} \; : \quad S', S \quad \mapsto \quad S'_{\tau i} \; \mathcal{T}_{\tau \tau' \tau''} \; \mathcal{W}_{ij} \; V(S)_{jk\tau'\tau''}} \tag{8}$$

where for notional simplicity we split $S'$ a color and stack dimension, and we have used Einstein summation convention with $i, j, k = 1, \ldots, 3$, and $V(S)$ is computed via the SVD. We introduce trainable weights $\mathcal{T}$ which are unconstrained an break the time-reversal symmetry of the expression equation 8; $\tau, \tau', \tau'' = 1, \ldots, N - 1$.

### 3.1 Theoretical Aspects

**Proposition 1** (CiL Symmetry Invariance)**.** *Our approach equation 7 is invariant under:*

- *right matrix multiplication by orthogonal $\mathcal{L}$ equation 5 if $\mathcal{W} = 1$ i.e. equal to the identity matrix - Category I -.*

- *by a one-parameter family of (special) orthogonal $\mathcal{L}$s equation 5 if $\mathcal{W} \in SO(3)$. The specific $\mathcal{L}$ depend on the details of weights in $\mathcal{W}$, see the appendix section (B.1) and (C.2.1) for exact relations - Category II -.*

- *by 3 (6 [4]) symmetric $\mathcal{L}_1, \mathcal{L}_2, \mathcal{L}_3$ equation 5 iff $\mathcal{W}$ is symmetric. The specific $\mathcal{L}$ depend on the details of weights in $\mathcal{W}$, see the proof section (B.1) and (C.2.1) for exact relations - Category III -.*

**Proposition 2** (rDMD Symmetry Invariance)**.** *The rDMD approach equation 6 is invariant under left and right matrix multiplication by orthogonal $\mathcal{L}$, $\mathcal{R}$ equation 5, respectively.*

---

[2]We refer to *3-channel as the input color-channels which are duplicated in our layer.*

[3]As for color-channels $V$ and $V_r$ are mostly identical we use them synonymously from here one.

[4]The expression in the parenthesis in proposition (1) refer to the case when we do not normalise V by its determinant. In the default case we use $V \to V/\det(V)$.

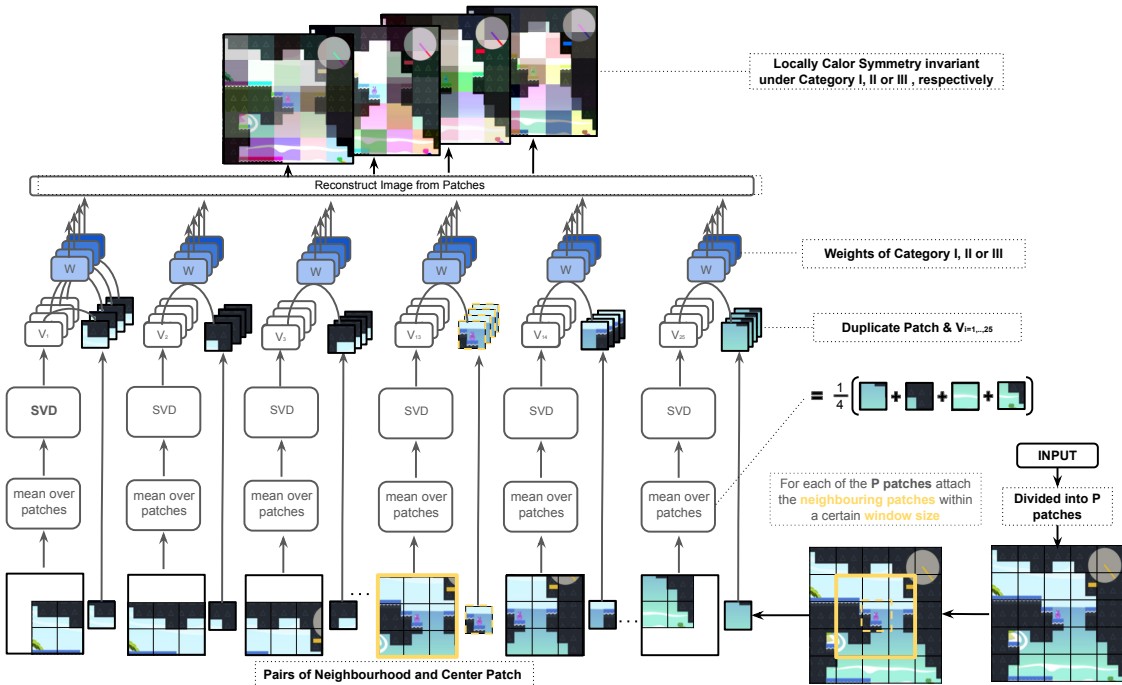

**Figure 3:** Architecture diagram of CiL for $N = 1$, i.e. no frame stacking.

**Controlling spatial and temporal symmetries of rDMD**. Among other symmetries, equation 3 and equation 6 admit permutation invariance and general rotation of pixels as long as the operation acts on each time step in the frame stack of input images alike, see Figure (4) (c) and (d). It is evident that those invariances admit no practical use case. It is this desirable to remove those symmetries while keeping the useful ones e.g. Figure (4) (b). In (Weissenbacher et al., 2024) recently a graph symmetric approach to symmetry breaking in pixel data was discussed. We define

$$\mathcal{K}_r^{break} = U_r^* \, K \, U_r = U_r^* \, \mathcal{G} \, S' \, \mathcal{T} \, V_r \, \Sigma_r^{-1} \quad . \tag{9}$$

where $\mathcal{G}$ is the graph matrix discussed in (Weissenbacher et al., 2024) and is such that $\begin{bmatrix} \mathcal{G} , \mathcal{R} \end{bmatrix} = \mathcal{G} \mathcal{R} - \mathcal{R} \mathcal{G} = 0$ i.e. they commute if $\mathcal{R}$ is a simple left right rotation or flip of the patch. In particular, for general permutation of pixels in the patch they do not commute. The matrices $\mathcal{G} \in \mathbb{R}^{n \times n}$, $\mathcal{T} \in \mathbb{R}^{N \times N}$ and $\mathcal{W} \in \mathbb{R}^{3 \times 3}$ contain trainable weights. While we do not directly use equation 9 involving a graph matrix in this work our formalism can be extended to include the latter.

**Proposition 3** (rDMD - Spatial Symmetry Breaking). *Eq. (9) is invariant under left and right matrix multiplication by orthogonal $\mathcal{L}$, $\mathcal{R}$ equation 5, which commute with $\mathcal{G}$. Be $\mathcal{G} \in \mathbb{R}^n \times \mathbb{R}^n$ is given by the symmetric graph-matrix, i.e. entries have shared weights if the distance between patches is the same. Then the symmetries of rotations, flips are preserved while the symmetry of undesirable general pixel permutations and orthogonal transformations is broken.*

**Proposition 4** (CiL-Stack Imposing Time-order). *By adding the weights $\mathcal{T} \in \mathbb{R}^{N \times N}$ of the frame stack version of CiL equation 8 one imposes a time-ordering of the frames, i.e. one generally breaks the orthogonal symmetries applied to the stack-dimension.*

### 3.2 On the importance of reducing an abundance of symmetry

In the previous section we have laid the theoretical basis of how specific symmetries of CiL may be broken. In this concluding section we informally highlight some of the findings of section 3.1. In Figure (4) we illustrate patch-wise and global transformation of spatial and color. We proof in the appendix C.2.4 that CiL is not invrant under regional color changes Figure (4) (a). This property is crucial to be able to identify color sensitive feature e.g. teh red vs green traffic light.

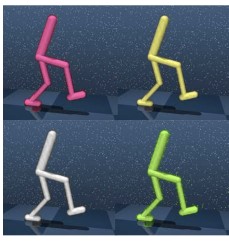 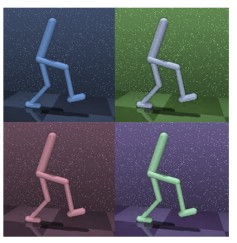 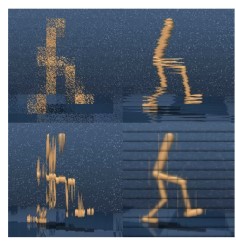 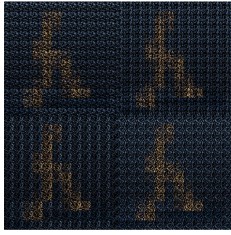

**(a)** Regional color changes. Broken by CiL, see appendix (C.2.4).

**(b)** Orthogonal color changes i.e. global application of $\mathcal{L}$.

**(c)** Patchwise pixel permutations and blurrs (spatial transformation).

**(d)** Patchwise orthogonal matrix multiplication by R.

**Figure 4:** Comparison of symmetries which aid learning (a) and (b) contrasted to those impeding the ability to learn crucial features (c) and (d).

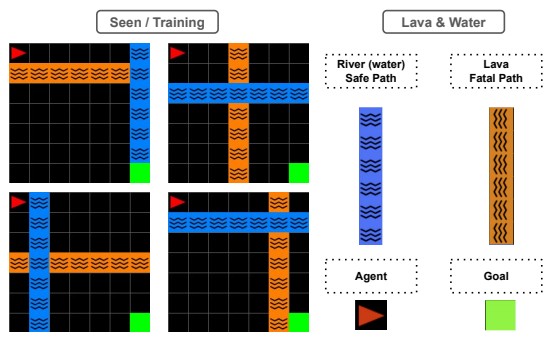

**(a)** Modified Lavacrossing, River (blue) are safe passage through Lava stream (orange).

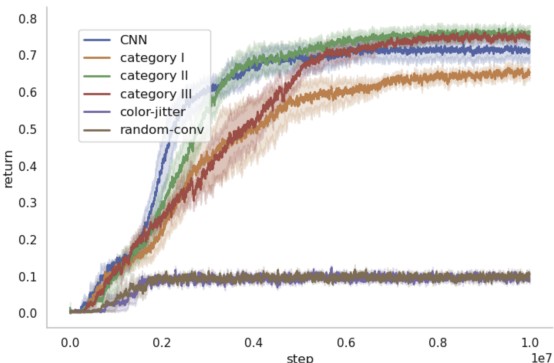

**(b)** Training rewards, averaged over 4 random seeds.

**Figure 5:** Modified lava-crossing environment overview and empirical evaluation of CiL. Comparison of CiL to Color-jitter and Random-conv data augmentation, the latter are detrimental for learning.

In Figure (4) (b) global color-transformations by distinct symmetric matrices $\mathcal{L}$ are shown. Global refer to the choice of the local neighborhood of SVD in Figure (3) to be the entire image, i.e. the same V is the shared across all patches.

In Figure (4) (c) and (d) we highlight patch-wise spatial transformation e.g. pixel-permutations and general orthogonal transformation. Those result in apparent non-useful symmetry properties. A model with those too abundant symmetries fails to learn relevant features necessary for the agent to navigate the environment.

In Figure 3, the different patch colors are for illustration only. The patches are not independent; the same color invariance is learned for each feature across all patches, as CiL's weights are shared among them. One might wonder if patch-wise color augmentation could yield similar results to CiL. However, empirical tests on Minigrid show that patch-wise augmentations, like global color augmentations, also fail to learn effectively. Additionally, to achieve the same invariance properties as CiL, the model would require an impractical number of patch-wise augmentations, namely: (#local color augmentations)$^{\#\text{patches}}$.

# 4 Empirical evaluation

## 4.1 Modified Minigird Lavacrossing to Test Color-sensitivity & Generalisation

The LavaCrossing environment, a standard in the MiniGrid toolkit (Chevalier-Boisvert et al., 2019), requires agents to navigate to a goal (green square) without falling into lava (orange squares). We modify the environment by adding:

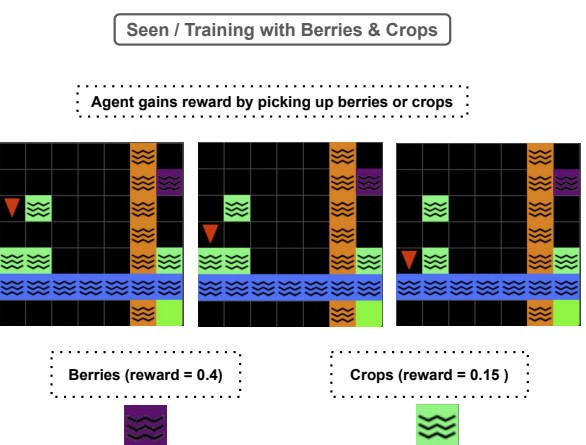

**Figure 6:** Modified Lava-crossing with additional option for the agent to collect rewards by harvesting berries (purple) and crops (green).

1. a safe river (water) to the environment which constitutes the only safe pathway through the lava stream, see Figure (5a). Thus the agent must learn to take a "swim" in the river which is a color sensitive choice as the lava needs to be avoided.

2. optionally to point (1) we introduce berry-fields and crop-fields see Figure (6) upon collecting them the agent receives a additional rewards.

Thus depending on the colors blue and orange of river vs. lava (purple and green of berries and crops) see Figure (6) opposing actions need to be taken by the agent to successfully reach the green goal. In other words, this modified environment is an ideal test-ground for an agent's ability of developing color-sensitivity, see Figure (10).

Although MiniGrid setups are usually partially observable, we adjust ours for full observability and increase the default observation size from $9 \times 9$ to $14 \times 14$ pixels. In particular, we render the environment and subsequently down-scale it.The experiments with deep Q-learning (IMPALA (Espeholt et al., 2018)) focus on difficulty level 1 of our modified Lava-crossing environment.

We evaluated the CNN baseline against configurations using color-jitter or random-conv data augmentation. These methods were unsuccessful in learning the desired behavior as they hindered the agent's capacity for color-sensitive decision-making. In contrast, when we incorporated CiL with the CNN for color-symmetry categories I, II, and III, learning proceeded without obstruction. Refer to Figure (5b) for the training rewards. Our findings suggest that not only does CiL achieve inherent generalization, but it also equips the model with color-sensitive decision-making skills.

Furthermore, Figure (5b) and (7a) illustrates the enhanced performance and sample efficiency of CiL + CNN, for both environment versions, respectively. Moreover, Figure (7b) shows improved color generalisation of CiL. Notably, this improvement is observed without any adjustments to the hyper-parameters of the RL algorithm.

Next, we extend the empirical results on our custom LavaCrossing environment and present evaluations of point (2) below. We modify the environment[5] by adding we introduce berry-fields as well as crop-fields upon collecting them the agent receives additional rewards.

In evaluating the performance of CiL on the environment option (2), which includes both berries and crops, we observe results that are consistent with those obtained in environment option (1). This can be seen in Figure (7a), which presents a visual representation of the findings.

---

[5]The LavaCrossing environment, a standard in the MiniGrid toolkit (Chevalier-Boisvert et al., 2019), requires agents to navigate to a goal (green square) without falling into lava (orange squares).

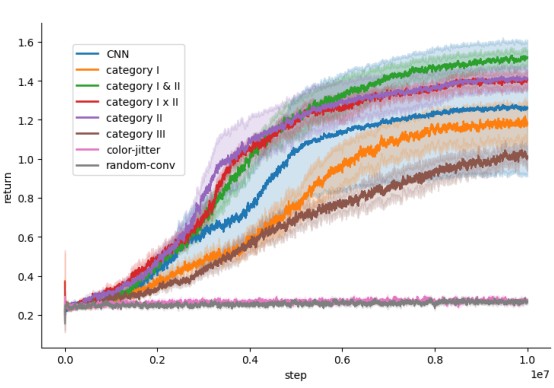

**(a)** Training rewards, averaged over 4 random seeds.

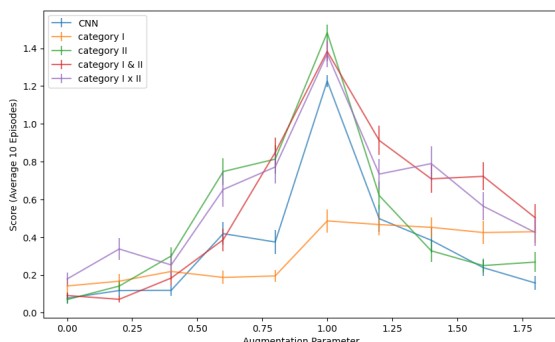

**(b)** Color Generalisation: rewards, averaged over 4 random seeds and 50 evaluation episodes; we plot the standard error bars, respectively. Augmentation parameter are from left to right as in Figure 6.

**Figure 7:** Empirical evaluation of CiL on modified lava-crossing environment (2) with berries and crops. Comparison of CiL to color-jitter and random-convolution data-augmentation, the latter are detrimental for learning. Comparison of Categories I vs. II, with feature-concatenated as the input of the CNN - *I&II* - and batch-concatenated i.e. used simultaneously - *I × II* -. Both Figure 7a and 7b show mean and standard error.

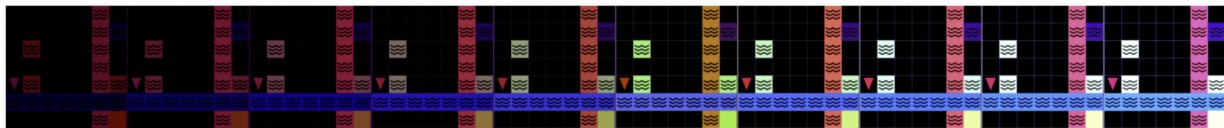

**Figure 8:** Modified Lava-crossing color variations for Figure 7b visualizing the augmentation parameter from left to right [0, 1.8]

When data-augmentation is applied, we notice a detrimental effect on the training process. This is in line with our observations from environment option (1), suggesting that data-augmentation might not be beneficial for this specific task.

Furthermore, we observe that the CiL category II demonstrates superior performance when compared to the CNN baseline when no data augmentation is applied. This performance advantage is more pronounced in environment option (2) than what was observed in environment option (1), as depicted in Figure (5b). This indicates that the benefits of using CiL category II become more evident when the complexity of the environment increases, showcasing its robustness and effectiveness in more challenging scenarios.

We evaluate our trained models on a color varied version of the environment, see Figure (8); comparing CNN to CNN + CiL for categories I,II, and II, respectively. We find improved generalisation of CiL, see Figure (7b).

## 4.2 Procgen Benchmark & Distracted Deepmind Control suite

We evaluate our model on both the Procgen generalization benchmark (Cobbe et al., 2020) which consists of 16 procedurally generated environments with visual observations; as well as the Deepmind Control suite (DMControl) (Tassa et al., 2018) with additional visual background video distraction (Hansen & Wang, 2021).

The Procgen benchmark consists of sixteen procedurally generated games. Each game corresponds to a distribution of partially observable Markov decision processes (POMDPs) $q(m)$, and each level of a game corresponds to a POMDP sampled from that game's distribution $m \sim q$. The POMDP $m$ is determined by the seed (i.e. integer) used to generate the corresponding level. Following the setup from (Cobbe et al., 2020), agents are trained on a fixed set of n = 200 levels (generated using seeds from 1 to 200) and tested on the full distribution of levels (generated by sampling seeds uniformly at random from all computer

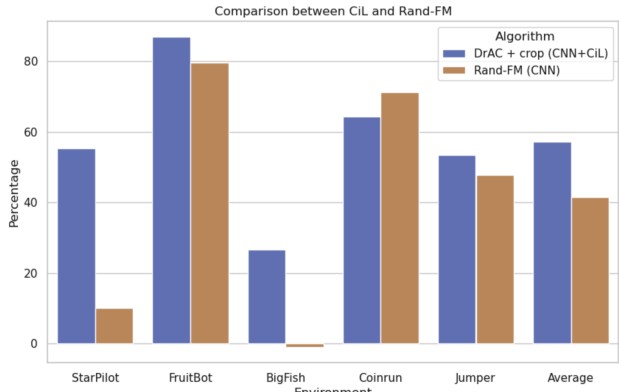

**Figure 9:** Comparison of CiL to random-conv augmentation on top of PPO backbone.

integers).We evaluate our method with PPO/DrAC (Schulman et al., 2017; Raileanu et al., 2020) with added crop augmentation.

Moreover, we evaluate our method with SAC (Haarnoja et al., 2018) with standard hyper-parameter settings, and simply add CiL as a preliminary layer. To evaluate generalization of our method and baselines, we test methods under challenging distribution shifts from by adding background videos (Hansen & Wang, 2021). We provide a proof-of-concept for CiL's ability to generalize to background videos in DMControl. This successful test is significant to us, as SVD/DMD can be highly sensitive to dynamic changes introduced by background videos. The DMControl test demonstrates CiL's effective generalization to new data distributions relevant for real-world tasks, rather than aiming to outperform state-of-the-art (SOTA) methods.

Based on the Procgen and DM-Control (DMC) results in Tables (1) and (3), and the Procgen ablations in Table (2), we can draw three main conclusions:

- First, CiL can scale to Procgen and DMC tasks.

- Second, CiL may be simply added as a preliminary layer to existing architectures; across a variety of domains and tasks and does not require any hyper-parameter tuning to achieve comparable - or improved - generalisation performance. The performance gains are significant for environments where color-sensitivity is required e.g. StarPilot and BigFish, in particular compared to rand-conv and color-jitter data augmentations.

- Third, the ablation study of CiL on Procgen finds that its performance is consistent across a wide range of settings.

In Figure (9), we juxtapose CiL with random-conv augmentations. Specifically for the Starpilot and Bigfish environments, the use of random-conv (Rand-FM) hinders learning. In contrast, CiL offers advantages over both PPO and DrAC+ Crop, especially in these environments. This aligns with our main results Table 1 where random-conv (as well as color-jitter) are added to DrAC. The training curves show comparable sample-efficiency of CiL, see Figure (11).

The primary conclusion of the paper is that a mixture of different categories yields the best performance. The general principle is that by incorporating all categories—each with distinct inductive biases—the network is better equipped to select the most appropriate features.

### 4.3 Limitations

CiL is CPU extensive by employing SVD as existing GPU implementation of SVD barely bring speed-ups, and rather lead to bottle-necks. Repeating input data (refer to Figure (3)) to assign different symmetry weights across channels can be demanding in terms of memory and computation. Particularly, using SVD locally in CiL increases runtime by a factor of 3-5 compared to the CNN baseline. This work serves as a

**Table 1:** We experiment on the Procgen environments. We compare to the PPO baseline and to Rand-FM (Lee et al., 2020) with a ResNet architecture. We report min-max normalised scores averaged over the testing steps 23M-25M, and 4 seeds, respectively. We use DrAC + crop augmentation for our model in which we use a preliminary CiL and the same subsequent ResNet. The CiL used local neighbourhood of 5 patches both vertically and horizontally and categories I, II and III. Both random convolution and color-jitter augmentations are added (for Conv&Jitter with equal likelihood). **The standard error for CNN+CiL as well as baseline is $< \approx 0.2\%$ for each game individually.**

| Procgen | | | | | | | |
|---|---|---|---|---|---|---|---|
| **Algorithm** | **Model** | **StarPilot** | **FruitBot** | **BigFish** | **Coinrun** | **Jumper** | **Average** |
| PPO | CNN | 35.6% | 85.9% | 7.8% | 65.4% | 53.3% | 49.6% |
| DrAC (Crop) | CNN | 50.0% | 86.5% | 21.6 % | 64.5% | 52.4% | 55.0% |
| DrAC (Crop+RandConv) | CNN | 47.3% | 84.8% | 14.2% | 64.9% | 56.5% | 53.5% |
| DrAC (Crop+ColorJitter) | CNN | 52.2% | 85.9% | 13.4% | 64.9% | 53.8% | 54.1% |
| DrAC (Crop+Conv&Jitter) | CNN | 44.8% | 84.0% | 9.4% | 65.0% | 56.2% | 51.9% |
| DrAC (Crop) | CNN+**CiL** | **55.4%** | 86.9% | **26.6%** | 64.4% | 53.4% | **57.3%** |
| Rand-FM | CNN | 10.1% | 79.7% | -1.1% | 71.2% | 47.8% | 41.5% |

**Table 2:** Procgen ablation study comparing categories I, II, and III as well as different local neighborhood sizes. The superscript refers to the weight categories used in CiL. The subscript $_1$, $_3$, $_5$, $_*$ refer to the size of the local neighborhood used to compute the SVD; where $*$ denotes the global CiL, i.e., the mean is computed over patches of the entire image.

| Procgen Task | $\text{CiL}_*^{I\,\&\,II}$ | $\text{CiL}_*^{I\,\&\,III}$ | $\text{CiL}_*^{I-III}$ | $\text{CiL}_1^{I-III}$ | $\text{CiL}_3^{I-III}$ | $\text{CiL}_5^{I-III}$ |
|---|---|---|---|---|---|---|
| StarPilot | 53.0% | 44.8% | 47.8% | 51.2% | 50.7% | **55.4%** |
| FruitBot | 86.6% | 86.3% | 88.0% | 86.7% | 87.6% | **86.9%** |
| BigFish | 21.6% | 14.0% | 21.3% | 11.7% | 23.7% | **26.6%** |
| Coinrun | 63.5% | 60.8% | 53.7% | 64.2% | 64.9% | 64.4% |
| Jumper | 53.0% | 46.6% | 48.3% | 54.7% | 54.4% | 53.4% |
| Average | 55.5% | 50.5% | 51.8% | 53.7% | 56.3% | **57.3%** |

**Table 3:** DM-control with background video distraction as well as color changes. We evaluate the trained models after 100k steps over 10 episodes, respectively, and report the mean and standard deviation over 2 seeds.

| DMControl | | | |
|---|---|---|---|
| **Domain-Task** | **Distraction** | **CNN** | $\textbf{CiL}_5^{I\,\&\,II}$ |
| Walker-Walk | video-easy | $493.1 \pm 125.0$ | $\textbf{497.9} \pm 118.8$ |
| | video-hard | $137.5 \pm 72.6$ | $112.6 \pm 48.1$ |
| Cheetah-Run | video-easy | $167.3 \pm 99.9$ | $\textbf{169.2} \pm 58.5$ |
| | video-hard | $41.1 \pm 26.1$ | $\textbf{61.8} \pm 44.6$ |

proof of principle, highlighting a new research direction with scope for future technical enhancements. We effectively reduced overhead by incorporating the SVD of the input frame-stack into the replay-buffer, as demonstrated in our DMC experiments.

## 5 Related Work

Symmetry is a prevalent implicit approach in deep learning for designing neural networks with established equivariances and invariances. The literature on symmetries in Vision Transformers (ViTs) (Fuchs et al., 2020; Romero & Cordonnier, 2021) is relatively limited compared to CNNs (Zhang & Sejnowski, 1988; LeCun et al., 1989; Zhang, 1990), recurrent neural networks (Rumelhart et al., 1986; Hochreiter & Schmidhuber, 1997), graph neural networks (Maron et al., 2019; Satorras et al., 2021), and capsule networks (Sabour et al., 2017). Permutation invariance in ViTs and attention mechanisms has been examined in (Lee et al., 2019), demonstrating improved out-of-distribution generalization in RL from pixel data (Tang & Ha, 2021). CiL may be added on top of ViTs as well as CNNs.

Conventionally sample efficiency is enhanced by data augmentation (Krizhevsky et al., 2012). Simple image augmentations, such as random crop (Laskin et al., 2020) or shift (Yarats et al., 2021a; 2022), can improve RL generalisation performance; in particular when combined with contrastive learning (Agarwal et al., 2021). CiL is complementary to data-augmentation.

3D rotations, described by the $SO(3)$ group, have found their place in various areas of vision-based AI tasks, however applied on the spatial and not color-features. From 3D medical imaging (Vieweg et al., 2015) over 3D object recognition (Qi et al., 2017; Coors et al., 2018) to transferring neural networks from simulations to real-world robotics (Tobin et al., 2017). Furthermore, deep learning on Lie Groups, encompassing $SO(3)$, has evolved with architectures like the Spherical CNNs (Cohen et al., 2018), inherently managing rotations.

Symmetry-based representation learning (Balaraman & Andrew, 2004; van der Pol et al., 2020; Higgins et al., 2018) refers to the study of symmetries of the environment manifested in the latent representation and was extended to environmental interactions in (Caselles-Dupré et al., 2019). In (Weissenbacher et al., 2022), symmetries of the dynamics are inferred in a self-supervised manner. More recent works equivariant methods in RL include (van der Pol & Welling, 2019; Wang & Walters, 2022). These approaches are mostly complimentary to CiL.

## 6 Conclusions

In the context of reinforcement learning, we introduced CiL, a neural network layer capable of self-supervised adaptation to various color symmetries. Empirical evaluations show that CiL outperforms both random-conv and color-jitter data augmentations in environments where color sensitivity is pivotal. Furthermore, CiL maintains performance in other environments without necessitating hyperparameter tuning for the associated RL algorithms.

## Impact Statement

This paper presents work whose goal is to advance the field of Machine Learning. There are many potential societal consequences of our work. In particular, while improved color-sensitivity while maintaining color-generalisation is desirable for many tasks , there are risks, such as an agent learning to make decisions based on an individual's skin color. However, we conclude that such a behavior would likely arise from biases in the training dataset or environment, rather than from our method itself.

## Acknowledgments

Matthias Weissenbacher would like express his gratitude to Rishabh Agarwal (Deepmind) for initial collaboration and helpful comments on the draft. And, we would like to thank Y. Nishimura for technical support. This work was supported by JSPS KAKENHI Grant Number JP22H00516 and JP22H05106 JST CREST Grant Number JPMJCR1913.

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

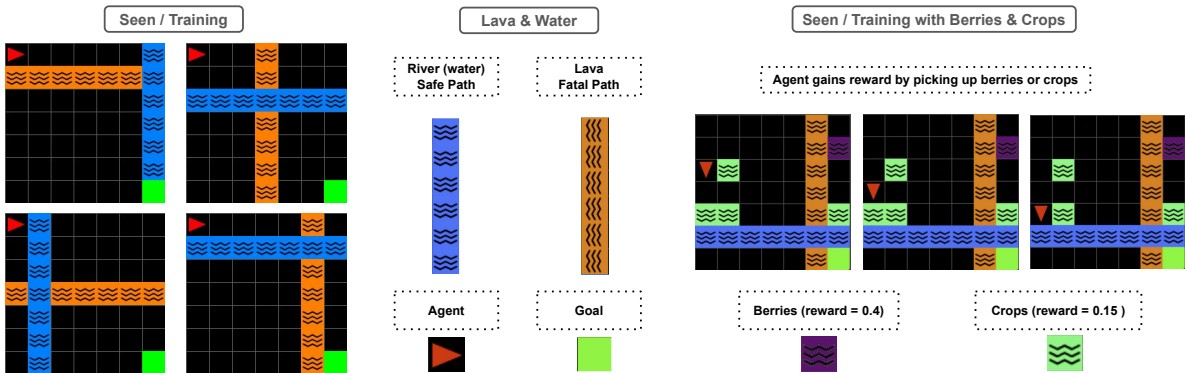

**Figure 10:** Modified Lava-crossing with additional option for the agent to collect rewards by harvesting berries (purple) and crops (green).

## A    Modified Lavacrossing Environment - Add-On

In this section we extend the empirical results on our custom LavaCrossing environment and present evaluations of point (2) below. We modify the environment[6] by adding:

1. a safe river (water) to the environment which constitutes the only safe pathway through the lava stream, see Figure (5a). Thus the agent must learn to take a "swim" in the river which is a color sensitive choice as the lava needs to be avoided.

2. optionally to point (1) we introduce berry-fields as well as crop-fields upon collecting them the agent receives additional rewards.

In evaluating the performance of CiL on the environment option (2), which includes both berries and crops, we observe results that are consistent with those obtained in environment option (1).

When data-augmentation is applied, we notice a detrimental effect on the training process. This is in line with our observations from environment option (1), suggesting that data-augmentation might not be beneficial for this specific task.

Furthermore, we observe that the CiL category II demonstrates superior performance when compared to the CNN baseline when no data augmentation is applied. This performance advantage is more pronounced in environment option (2) than what was observed in environment option (1), as depicted in Figure (5b). This indicates that the benefits of using CiL category II become more evident when the complexity of the environment increases, showcasing its robustness and effectiveness in more challenging scenarios.

### A.1    Color Generalisation

We evaluate our trained models on a color varied version of the environment, see Figure (10); comparing CNN to CNN + CiL for categories I,II, and II, respectively. We find improved generalisation of CiL, see Figure (7b).

---

[6]The LavaCrossing environment, a standard in the MiniGrid toolkit (Chevalier-Boisvert et al., 2019), requires agents to navigate to a goal (green square) without falling into lava (orange squares).

$$\mathcal{W} = \begin{bmatrix} \cos\phi\cos\theta & \cos\phi\sin\theta\sin\psi - \sin\phi\cos\psi & \cos\phi\sin\theta\cos\psi + \sin\phi\sin\psi \\ \sin\phi\cos\theta & \sin\phi\sin\theta\sin\psi + \cos\phi\cos\psi & \sin\phi\sin\theta\cos\psi - \cos\phi\sin\psi \\ -\sin\theta & \cos\theta\sin\psi & \cos\theta\cos\psi \end{bmatrix} , \tag{10}$$

## B   Technical Details

### B.1   Implementation of Symmetry Groups

#### B.1.1   SO(3) group - Category II

The group $SO(3)$ represents the group of rotations in three dimensions, and any rotation in this group can be represented using three Euler angles. Euler angles are three angles introduced by Leonhard Euler to describe the orientation of a body/vector. They can describe arbitrary rotations in three dimensions. The three rotations are often referred to as roll, pitch, and yaw, especially in the context of aviation and robotics.

Any element of $\rho \in SO(3)$, i.e., any rotation matrix, can be expressed in terms of Euler angles. There are multiple sequences of axes about which the rotations can take place (like "XYZ", where the rotation is first about X, then about Y, and lastly about Z), and the choice of sequence can change the specific angles.

For a rotation in 3D space, a common sequence is the "Z-Y-X" sequence (yaw, pitch, and roll, respectively):

1. A rotation $\phi$ (yaw) about the Z-axis.

2. Followed by a rotation $\theta$ (pitch) about the new Y-axis.

3. Followed by a rotation $\psi$ (roll) about the new X-axis.

Given these angles, the overall rotation matrix $\rho$ in the "Z-Y-X" sequence can be given as $\rho = \rho_x(\psi)\rho_y(\theta)\rho_z(\phi)$ where:

$$\rho_z(\phi) = \begin{bmatrix} \cos\phi & -\sin\phi & 0 \\ \sin\phi & \cos\phi & 0 \\ 0 & 0 & 1 \end{bmatrix}$$

$$\rho_y(\theta) = \begin{bmatrix} \cos\theta & 0 & \sin\theta \\ 0 & 1 & 0 \\ -\sin\theta & 0 & \cos\theta \end{bmatrix}$$

$$\rho_x(\psi) = \begin{bmatrix} 1 & 0 & 0 \\ 0 & \cos\psi & -\sin\psi \\ 0 & \sin\psi & \cos\psi \end{bmatrix}$$

When expanded, the combined rotation matrix written as in the weight notation $\mathcal{W}$ is:[7] with the bounds for the Euler angles:

- Yaw ($\phi$): $0 \leq \phi < 2\pi$

- Pitch ($\theta$): $-\frac{\pi}{2} \leq \theta \leq \frac{\pi}{2}$

- Roll ($\psi$): $0 \leq \psi < 2\pi$

We employ the definition equation 10 to incorporate SO(3) rotation of color-space i.e. category II, where the Euler angles are given by three trainable weights. the bounds are enforces by using the Tangent-Hyperbolic function e.g. $0 \leq \phi < 2\pi$ by $\phi = \pi + \pi * \text{Tanh}(\theta)$, where $\theta$ is a trainable parameter.

---

[7]Note: While Euler angles are intuitive, they are not without problems. The most notorious one is "gimbal lock," where you lose one degree of freedom, and it's not possible to represent all 3D orientations. This is why other representations like quaternions are sometimes preferred in applications like computer graphics and robotics.

### B.1.2 Symmetric Weight Matrices - Category III

For category III the weight matrices $\mathcal{W}$ are symmetric.[8] This is accomplished by simple defining a $3 \times 3$ parameter matrix $\Theta$ and symmetrise i.e add the transpose as

$$\mathcal{W} = \tfrac{1}{2}\left(\text{softmax}(\Theta) + \text{softmax}(\Theta)^T\right) \ . \tag{11}$$

We add the softmax function is added to restrict the values of the weights.

Note that symmetric matrices do not form a group. The product of two symmetric matrices is symmetric if and only if the matrices commute, meaning

$$AB = BA.$$

If matrices $A$ and $B$ are symmetric, then:

$$(AB)^T = B^T A^T$$

Given that $A$ and $B$ are symmetric:

$$B^T = B$$
$$A^T = A$$

So,

$$(AB)^T = BA$$

For $AB$ to be symmetric, $(AB)^T$ must equal $AB$, so

$$AB = BA.$$

### B.1.3 Numerical Invariance: `torch.svd`

In our implementation, we have opted for `torch.svd` over `torch.linalg.svd` for computing the singular value decomposition (SVD). Our extensive testing indicates that `torch.svd` offers superior stability in maintaining invariance under orthogonal transformations of the input, denoted as $R$ and $L$.

However, it is important to highlight that numerical algorithms like SVD are prone to challenges associated with sign ambiguity. This means that the columns of the output matrices can undergo multiplication by $\pm 1$. To tackle this issue and achieve consistent results, we employ `torch.abs` on the results of equations 7 and 8. This step is imperative to address the sign ambiguity introduced by `torch.svd`, ensuring that the final output is consistent and unaffected by potential sign changes.

In essence, the adoption of `torch.svd` and the application of `torch.abs` contribute to the stability and reliability of our implementation, especially in scenarios involving orthogonal transformations of the input.

### B.2 Normalized Rewards in Procgen

See Figure (11) for test curves or Procgen during training.

To compute normalized scores, we leverage a simple yet effective rescaling technique. By using the observed minimum and maximum rewards in the dataset, we ensure that the scores are bounded between 0 and 1. Specifically, given a reward value $rwd$, the normalized score is calculated using the formula:

$$\text{normalized score} = \frac{\text{reward} - \text{minimal reward}}{\text{maximal reward} - \text{minimal reward}}$$

---

[8]We incorporate symmetric matrices in our work which is not to be confused with a related concept to orthogonal groups the symmetric group. The symmetric group on three elements, often denoted $S_3$, is a mathematical structure that encapsulates all possible permutations of three distinct objects. This group has 3! (i.e., 6) distinct elements, each representing a unique permutation. When considering matrices, a natural representation of $S_3$ is as $3 \times 3$ permutation matrices where each matrix has a single "1" in every row and column, with all other entries being "0". Each of these matrices represents a permutation of the standard basis vectors in $\mathbb{R}^3$. All permutations are orthogonal thus $S_3$ is a sub-group of SO(3).

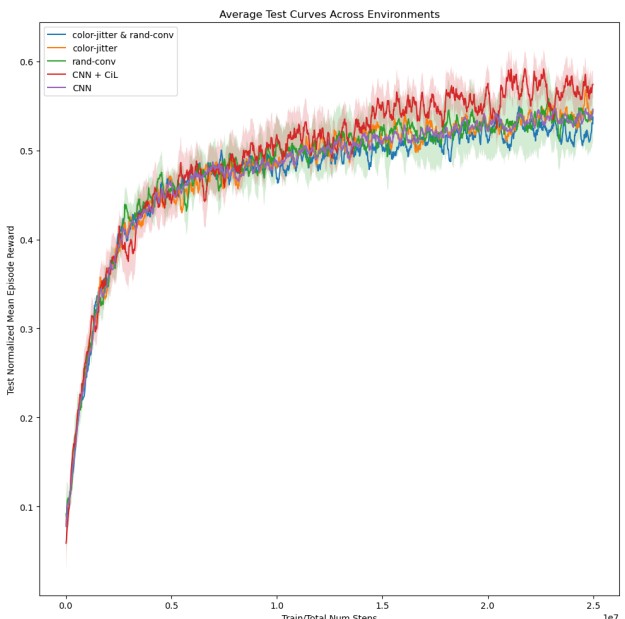

**Figure 11:** Sample-efficiency of CiL: Procgen testing curves during DrAC training of baselines and CiL, see Table 1. We average over all 5 environments and 4 seeds. We show the mean and the mean of the standard error of the individual tasks.

| Game | Min | Max |
|---|---|---|
| BigFish | 1 | 40 |
| StarPilot | 2.5 | 64 |
| FruitBot | -1.5 | 32.4 |
| CoinRun | 0.5 | 13 |
| Jumper | 1 | 10 |

**Table 4:** Min-max rewards for Procgen environments used to compute normalized scores in Table (1) and (2).

Here, minimal reward represents the smallest observed reward in the dataset, and maximal reward represents the largest observed reward. This normalization approach ensures that rewards are set relative to their observed range, providing a standardized perspective on their values.

All experiments were performed on NVIDIA GPU A-100 or V-100.

### B.3 Hyperparameters

We summarize the hyperparameter choices in Table (5).

## C Proofs of Invariance

The goal of this section is to prove the invariance of the quantities defined in equations (3),(7),(6). We start by giving some necessary definitions and proving some preliminary results in section (C.1).

### C.1 Preliminary steps

Given $n, N > 0$, we define the following subset of $\mathbb{R}^{n \times N} \times \mathbb{R}^{n \times N} \times \mathbb{R}^{N \times N} \times \mathbb{R}^{N \times N}$

$$\mathfrak{T}_{n,N} := \{(S, S', \Lambda, U) | S' S'^{\dagger}]$$

**Table 5:** Architecture and hyper-parameter choices for CiL on Procgen, DMControl, Minigrid based on (Raileanu et al., 2020), (Hansen & Wang, 2021), and (Jiang et al., 2021), respectively. Channels refer to the category channels. We use the algorithms in the code-base without any hyper-parameter changes except for reduction of hidden-dim of the actor-critic networks to 64. The patch size follows the convention in Vision Transformers; for a 64x64 pixel input, we use 8x8 patches. In Figure 3, the neighborhood sizes are chosen as odd numbers (3, 5) to ensure a central patch with an equal number of neighboring patches on all sides.

| Suite | Model | Patch Size | CiL Channels [I,II,III] | Activation | Hidden Dim. |
|---|---|---|---|---|---|
| **Procgen** (*Raileanu et al.*, 2020) | *CNN* | 3 ResNet ((Raileanu et al., 2020)) | [0,0,0] | N/A | 64 |
| | $\text{CiL}_*^{I\&II}$ | 8x8 pixel (64 patches) | [1,20,0] | selu | 64 |
| | $\text{CiL}_*^{I\&III}$ | 8x8 pixel (64 patches) | [1,0,20] | selu | 64 |
| | $\text{CiL}_*^{I-III}$ | 8x8 pixel (64 patches) | [1,10,10] | selu | 64 |
| | $\text{CiL}_1^{I-III}$ | 8x8 pixel (64 patches) | [1,10,10] | selu | 64 |
| | $\text{CiL}_3^{I-III}$ | 8x8 pixel (64 patches) | [1,10,10] | selu | 64 |
| | $\text{CiL}_5^{I-III}$ | 8x8 pixel (64 patches) | [1,10,10] | selu | 64 |
| **DMControl** (*Hansen&Wang*, 2021) | $\text{CiL}_5^{I\&II}$ | 8x8 pixel (144 patches) | [1,1,0] | selu | 64 |
| **Minigrid** (*Jiang et al.*, 2021) | $\text{CiL}_*^{I}$ | 7x7 pixel (4 patches) | [1,0,0] | selu | 32 |
| | $\text{CiL}_*^{II}$ | 7x7 pixel (4 patches) | [0,1,0] | selu | 32 |
| | $\text{CiL}_*^{III}$ | 7x7 pixel (4 patches) | [0,0,1] | selu | 32 |

is diagonalizable and

$$(\Lambda, U) \text{ is an eigendecomposition of } S'S^\dagger\}.$$

Let $R \in \mathbb{R}^{n \times n}$, $L \in \mathbb{R}^{N \times N}$ be orthogonal matrices; their orthogonality implies that

$$(RS')(RS)^\dagger = RS'S^\dagger R^{-1} \tag{12}$$

and

$$(S'L)(SL)^\dagger = S'LL^{-1}S^\dagger = S'S^\dagger \tag{13}$$

Therefore we arrive at the following lemma:

**Lemma 1.** *Let $(\Lambda, U)$ be an eigen-decomposition of $S'S^\dagger$ and let $R \in \mathbb{R}^{n \times n}$, $L \in \mathbb{R}^{N \times N}$ be orthogonal matrices. Then*

1. *The pair $(\Lambda, RU)$ is an eigen-decomposition of $(RS')(RS)^\dagger$.*

2. *The pair $(\Lambda, U)$ is an eigen-decomposition of $(S'L)(SL)^\dagger$.*

*Proof.* For the first part, using (12), we have

$$(RS')(RS)^\dagger(RU) = RS'S^\dagger R^{-1}(RU) = R(S'S^\dagger)U,$$

which is equal to $R(U\Lambda) = (RU)\Lambda$ by the hypothesis. The second part follows immediately by (13) and the hypothesis. $\square$

The above lemma allows us to define a left action of the group of orthogonal $n \times n$ matrices $O_n$ on $\mathfrak{T}_{n,N}$

$$\Phi_{n,N} : O_n \times \mathfrak{T}_{n,N} \to \mathfrak{T}_{n,N}$$
$$\big(R, (S, S', \Lambda, U)\big) \mapsto (RS, RS', \Lambda, RU)$$

as well as a right action of the group of orthogonal $N \times N$ matrices $O_N$ on $\mathfrak{T}_{n,N}$

$$\Psi_{n,N} : O_N \times \mathfrak{T}_{n,N} \to \mathfrak{T}_{n,N}$$
$$\big(L, (S, S', \Lambda, U)\big) \mapsto (SL^{-1}, S'L^{-1}, \Lambda, U).$$

We further define the functions

$$\chi_1 : \mathfrak{T}_{n,N} \to \mathbb{R}^{n \times n}$$
$$(S, S', \Lambda, U) \mapsto \mathrm{diag}\big( U^T S' S^\dagger U \big)$$

and

$$\chi_2 : \mathfrak{T}_{n,N} \to \mathbb{R}^{n \times N}$$
$$(S, S', \Lambda, U) \mapsto U^T S'$$

**Proposition 5.** *1. The function $\chi_1$ is invariant with respect to the group actions $\Phi_{n,N}$ and $\Psi_{n,N}$.*

*2. The function $\chi_2$ is invariant with respect to the group action $\Phi_{n,N}$.*

*Proof.* 1. For any $(R, (S, S', \Lambda, U)) \in O_n \times \mathfrak{T}_{n,N}$, we have

$$
\begin{aligned}
\chi_1 \quad \circ \quad & \Phi_{n,N}\Big( \big( R, (S, S', \Lambda, U) \big) \Big) \\
= \quad & \chi_1\big( (RS, RS', \Lambda, RU) \big) \\
= \quad & \mathrm{diag}\big( (RU)^T RS' (RS)^\dagger (RU) \big) \\
= \quad & \mathrm{diag}\big( (U^T R^T)(RS')(S^\dagger R^{-1})(RU) \big) \\
= \quad & \mathrm{diag}( U^T S' S^\dagger U ) \\
= \quad & \chi_1\big( (S, S', \Lambda, U) \big)
\end{aligned}
\tag{14}
$$

where we used the fact that $R$ is orthogonal to write $(RS)^\dagger = S^\dagger R^{-1}$ and $R^T R = I$.

For any $\big( L, (S, S', \Lambda, U) \big) \in O_N \times \mathfrak{T}_{n,N}$, we have

$$
\begin{aligned}
\chi_1 \quad \circ \quad & \Psi_{n,N}\big( L, (S, S', \Lambda, U) \big) \\
= \quad & \chi_1\big( (SL^{-1}, S'L^{-1}, \Lambda, U) \big) \\
= \quad & \mathrm{diag}\big( U^T (S'L^{-1})(SL^{-1})^\dagger U \big) \\
= \quad & \mathrm{diag}\big( U^T (S'L^{-1})(LS^\dagger) U \big) \\
= \quad & \mathrm{diag}\big( U^T S' S^\dagger U \big) \\
= \quad & \chi_1\big( (S, S', \Lambda, U) \big)
\end{aligned}
\tag{15}
$$

$$\tag{16}$$

where we used the orthogonality of $L$ to write $(SL^{-1})^\dagger = LS^\dagger$.

2. For any $(R, (S, S', \Lambda, U)) \in O_n \times \mathfrak{T}_{n,N}$, we have

$$
\begin{aligned}
\chi_2 \quad \circ \quad \Phi_{n,N}&\Big(\big(R, (S, S', \Lambda, U)\big)\Big) \\
&= \chi_2\big((RS, RS', \Lambda, RU)\big) \\
&= (RU)^T (RS') \\
&= U^T R^T R S' \\
&= U^T S' \\
&= \chi_2\big((S, S', \Lambda, U)\big)
\end{aligned}
\tag{17}
$$

$\square$

A *singular value decomposition* $(U, \Sigma, V)$ of $S$ is a factorization

$$
S = U \Sigma V^*
$$

where $U, V$ are unitary matrices and $\Sigma$ is rectangular diagonal with non-negative entries on its diagonal. Given $S$, such a decomposition need not be unique, however the matrix $\Sigma$ is unique up to an ordering of its diagonal entries (singular values of $S$). We will consider singular value decompositions where the diagonal entries of $\Sigma$ appear in descending order. Then, for any $r > 0$ we define $U_r$ to be the matrix formed by the first $r$ columns of U (also known as the modes).

**Lemma 2.** *Let $(U, \Sigma, V)$ be a singular value decomposition of $S$ and let $R \in \mathbb{R}^{n \times n}$, $L \in \mathbb{R}^{N \times N}$ be orthogonal matrices. Then*

    *1. $(RU, \Sigma, V)$ is a singular value decomposition of $RS$.*

    *2. $(U, \Sigma, L^{-1}V)$ is a singular value decomposition of $SL$.*

*Proof.* By the hypothesis we have $S = U\Sigma V^*$. Therefore, $RS = (RU)\Sigma V^*$ and $RU$ is orthogonal. Further, $SL = U\Sigma V^* L = U\Sigma (L^{-1}V)^*$ and $L^{-1}V$ is orthogonal. $\square$

Next, we introduce the following subset of $\mathbb{R}^{n \times N} \times \mathbb{R}^{n \times N} \times \mathbb{R}^{n \times n} \times \mathbb{R}^{n \times N} \times \mathbb{R}^{N \times N}$

$$
\mathfrak{T}_{n,N}^{SVD} := \{(S, S', U, \Sigma, V) \mid (U, \Sigma, V)
$$

is a singular value decomposition of $S$}. By Lemma 2 we are able to define a left action of the group of orthogonal $n \times n$ matrices $O_n$ on $\mathfrak{T}_{n,N}^{SVD}$

$$
\Phi_{n,N}^{SVD} : O_n \times \mathfrak{T}_{n,N}^{SVD} \to \mathfrak{T}_{n,N}^{SVD}
$$
$$
\big(R, (S, S', U, \Sigma, V)\big) \mapsto (RS, RS', RU, \Sigma, V)
$$

as well as a right action of the group of orthogonal $N \times N$ matrices $O_N$ on $\mathfrak{T}_{n,N}^{SVD}$

$$
\Psi_{n,N}^{SVD} : O_N \times \mathfrak{T}_{n,N}^{SVD} \to \mathfrak{T}_{n,N}^{SVD}
$$
$$
\big(L, (S, S', U, \Sigma, V)\big) \mapsto (SL^{-1}, S'L^{-1}, U, \Sigma, LV).
$$

Further, we define the following function

$$
k_r : \mathfrak{T}_{n,N}^{SVD} \to \mathbb{R}^{r \times r}
$$
$$
(S, S', U, \Sigma, V) \mapsto U_r^* S' S^\dagger U_r
$$

**Proposition 6.** *The function $k_r$ is invariant with respect to the actions $\Phi_{n,N}^{SVD}$ and $\Psi_{n,N}^{SVD}$.*

*Proof.* We have

$$
\begin{aligned}
k_r \quad \circ \quad & \Phi_{n,N}^{SVD}\big(R, (S, S', U, \Sigma, V)\big) \\
= \quad & k_r(RS, RS', RU, \Sigma, V) \\
= \quad & (RU)_r^*(RS')(RS)^\dagger(RU)_r \\
= \quad & (RU_r)^*(RS')(RS)^\dagger(RU_r) \\
= \quad & (U_r^* R^*)(RS')(S^\dagger R^{-1})RU_r \\
= \quad & U_r^* S' S^\dagger U_r \\
= \quad & k_r(S, S', U, \Sigma, V)
\end{aligned}
\tag{18}
$$

and

$$
\begin{aligned}
k_r \quad \circ \quad & \Psi_{n,N}^{SVD}\big(L^{-1}, (S, S', U, \Sigma, V)\big) \\
= \quad & k_r(SL, S'L, U, \Sigma, L^{-1}V) \\
= \quad & U_r^*(S'L)(SL)^\dagger U_r \\
= \quad & U_r^*(S'L)(L^{-1}S^\dagger)U_r \\
= \quad & U_r^* S' S^\dagger U_r \\
= \quad & k_r(S, S', U, \Sigma, V)
\end{aligned}
\tag{19}
$$

$\square$

## C.2 Completion of proofs

To complete the proofs, as usual a 2D image is identified with a matrix whose entries correspond to pixel color intensities. In our current setup, we further identify matrices with vectors by stacking together consecutive rows of a matrix, that is

$$\Xi : \mathbb{R}^{m \times k} \to \mathbb{R}^{mk}$$

$$
\begin{bmatrix}
x_{11} & x_{12} & \cdots & x_{1k} \\
x_{21} & x_{22} & \cdots & x_{2k} \\
\vdots & \vdots & \ddots & \vdots \\
x_{m1} & x_{m2} & \cdots & x_{mk}
\end{bmatrix}
\mapsto
\begin{bmatrix}
x_{11} \\
x_{12} \\
\vdots \\
x_{1k} \\
x_{21} \\
x_{22} \\
\vdots \\
x_{2k} \\
\vdots \\
x_{m1} \\
x_{m2} \\
\vdots \\
x_{mk}
\end{bmatrix}
$$

**Remark:** A rotation and a flip are special cases of pixel permutations, which, under the above identification, correspond to left multiplication by a permutation matrix, hence an orthogonal matrix.

$$\text{CiL-Stack} \; : \;\; S'L, SL \;\;\; \mapsto \;\;\; S'_{\tau i} \, (L_1)_{ij} (L_2)_{\tau\tau'} \; \mathcal{T}_{\tau'\tau''} \; \mathcal{W}_{ij} \;\; (L_1^T)_{jj'} (L_2^T)_{\tau''\tau'''} \, V(S)_{j'k\tau'''} \quad\quad (20)$$

### C.2.1  Proof of Proposition (1) and Proposition (4)

We build on the results of section (C.1). Let us repeat proposition (1) below.

**Proposition 7** (CiL Symmetry Invariance). *Our approach equation 7 is invariant under:*

1. *right matrix multiplication by orthogonal $\mathcal{L}$ equation 5 if $\mathcal{W} = 1$ i.e. the identity matrix.*

2. *by a one-parameter family of (special) orthogonal $\mathcal{L}$s equation 5 if $\mathcal{W} \in SO(3)$. The specific $\mathcal{L}$ depend on the details of weights in $\mathcal{W}$, see the appendix section (B.1) for exact relations.*

3. *by 3 (6) symmetric $\mathcal{L}_1, \mathcal{L}_2, \mathcal{L}_3$, equation 5 iff $\mathcal{W}$ is symmetric. The specific $\mathcal{L}$ depend on the details of weights in $\mathcal{W}$, see section (B.1) for exact relations.*

The expression in the parenthesis in proposition (1) refer to the case when we do not normalise V by its determinant. In the default case we use $V \to V/det(V)$.

**Proof:** The point (1) above follows as a consequence of the discussions in section (C.1).

The proof of point (2) and (3) of proposition (1) builds on point (1) thereof, i.e. by adding respective weights $\mathcal{W}$ to preserve specific symmetries and break others. Firstly, for claim (2) note that symmetries $L$ are preserved if and only if they commute with $\mathcal{W} \in SO(3)$ i.e. $[\mathcal{W}, L] = \mathcal{W}L - L\mathcal{W} = 0$. Where $L$ acts on the input $S \to SL$ thus

$$\text{CiL} \; : \;\; SL \;\; \mapsto \;\; SL \, \mathcal{W} \, L^T V$$

as $L$ is orthogonal i.e. $LL^T = L^T L = 1$ the expression above is invariant if and only if the claim of commutativity between $W$ and L holds.

Note that any real orthogonal $3 \times 3$ matrix $W \in SO(3)$ may be decomposed as $W = U\Lambda U^{-1}$ with one real and a pair of two complex conjugate eigen-values all lying on the unit disk (Strang, 2003). Thus the real eigen-values are either $\{+1, -1\}$ and the complex ones are $\{a+ib, a-ib\}$, $a, b \in \mathbb{R}$ s.t. $a^2 + b^2 = 1$. Moreover, invertible normal matrices (such as $W$ and $L$) commute if they have a shared set of eigen-vectors, i.e. they share the same eigen-decomposition but for different eigen-values i.e. $L = U\Lambda_L U^{-1}$.

We normalize the network to symmetries with determinant of $\{+1\}$ by adding to the layer $V \to V/det(V)$. Moreover, due to the constraint of lying of the eigenvalues on the unit disk only one independent parameter remains in $\Lambda_L$ i.e. is not fixed by the constraint of commuting with $W$. This free parameter in choosing $\Lambda_L$ constitutes the one-parameter family of symmetries which is preserved - category II symmetry type-.

The argument for claim (3) of proposition (1) is analog to the one for claim (2). With the difference that now $W$ is a symmetric matrix. Any real symmetric matrix can be written as $W = U\Lambda U^T$ for an orthogonal $U$, which is referred to as the Spectral Theorem (Strang, 2003). Since $U$ is real the eigenvalues of $L$ are constraint to $\{+1, -1\}$ and the additional constrain of determinant one, results in the following four choices for $\Lambda_L$: $\text{diag}(+1, -1, -1), \text{diag}(-1, +1, -1), \text{diag}(-1, -1, +1), \text{diag}(1, 1, 1)$. The latter is the identity matrix. If we allow $L$'s with determinant of either $\{+1, -1\}$ then one obtains eight different choices for $L$. As we omit plus/minus the identity matrix this results in 3 (6) choices for $L$ - category III -.

This concludes the proof of proposition (1). $\qquad\square$

**Proof of proposition (4).**  Let an orthogonal transformation $L$ act on the input $S \to SL$ and $S' \to S'L$ one arrives at equation 20, where we use Einstein summation convention $i, j, k = 1, \ldots, 3$ and $\tau, \ldots, \tau''' = 1, \ldots, N$ and where $L_1, L_2$ are orthogonal matrices transforming the color and stack dimension, respectively. Any orthogonal L may be decomposed in $L_1, L_2$. The proof for $L_1$ is the same as for proposition (1). As we choose $\mathcal{T} \in \mathbb{R}^{N \times N}$ to be generic the symmetry w.r.t. the orthogonal $L_2$ is broken as a general matrix does

not admit an orthogonal matrix it commutes with. However, since the weights of $\mathcal{T}$ are unconstrained those are free to preserve symmetries in a self-supervised way.

This concludes the proof of proposition (4). □

### C.2.2 Invariance of equation (3); proof of proposition (2), proposition (3)

By equation (13), equation (3) is invariant under the right action of an orthogonal matrix, hence invariant under orthogonal color changes (Figure 4(b)). This implies the statements of proposition (1).

Proposition (3) is a corollary of the results of (Weissenbacher et al., 2024), by noting that the graph matrix commutes with orthogonal transformations R if and only if R preserves the graph structure of underlying pixels, up to mirror flips and rotations by 90°.

### C.2.3 Rotations, flips, patchwise pixel permutations and multiplications and invertible color changes

Let us discuss the invariance of equations (6). By the above Remark and Propositions 5 and 6, $\chi_1$, $\chi_2$ and $\mathcal{K}_r$ are invariant under rotations, flips, patchwise pixel permutations and, more generally, patchwise multiplications by an orthogonal matrix $R$. The functions $\chi_1$ and $k_r$ are also invariant with respect to invertible color changes.

Regional color changes require some additional analysis.

### C.2.4 Regional Color Changes

Let

$$S \;=\; (s_{t_1}, s_{t_1}, s_{t_1}, s_{t_2}, s_{t_2}, s_{t_2}, \ldots, s_{t_N}, s_{t_N}, s_{t_N})$$

Since $\chi_1$ is invariant under matrix row permutations (Proposition 5), we may assume without loss of generality that the color changes apply to a fixed number $k$ of rows in bottom of the matrix $S$, while the rest of the rows above them are unchanged. In other words, we can define the regional color change operation

$$reg_L : \mathbb{R}^{n \times 3N} \to \mathbb{R}^{n \times 3N}$$

$$S = \begin{bmatrix} S_1 \\ S_2 \end{bmatrix} \mapsto \begin{bmatrix} S_1 \\ S_2 \cdot L \end{bmatrix}$$

where the matrix $S_1 \in \mathbb{R}^{(n-k) \times 3N}$ is the upper part of $S$, the matrix $S_2 \in \mathbb{R}^{k \times 3N}$ is the lower part of $S$ and $L$ is an arbitrary diagonal $3N \times 3N$ matrix, which acts by scaling the RGB channels of the fixed regions in the sequence of images represented by matrix $S_2$ (the notation $\cdot$ here refers to ordinary matrix multiplication).

**Claim 1**. *The matrix of eigenvalues in equation 7 is in general, **not invariant** under $reg_L$.*
*Proof.* Let us choose matrices

$$S = \begin{bmatrix} S_1 \\ S_2 \end{bmatrix}, S' = \begin{bmatrix} S_1' \\ S_2' \end{bmatrix}$$

such that

1. $S_1$ and $S_2$ are square orthogonal matrices of the same size.

2. $diag(S_1' S_1^{-1}) \neq diag(S_2' S_2^{-1})$.

$$reg_L(S') \cdot (reg_L(S))^\dagger = \begin{bmatrix} S'_1 & 0 \\ 0 & S'_2 \end{bmatrix} \cdot \begin{bmatrix} (I+L^2)^{-1} & L \cdot (I+L^2)^{-1} \\ L \cdot (I+L^2)^{-1} & L^2 \cdot (I+L^2)^{-1} \end{bmatrix} \cdot \begin{bmatrix} S_1^{-1} & 0 \\ 0 & S_2^{-1} \end{bmatrix}$$

$$= \begin{bmatrix} S'_1(I+L^2)^{-1}S_1^{-1} & S'_1 L \cdot (I+L^2)^{-1}S_2^{-1} \\ \\ S'_2 L \cdot (I+L^2)^{-1}S_1^{-1} & S'_2 L^2 \cdot (I+L^2)^{-1}S_2^{-1} \end{bmatrix}$$

$$\tag{22}$$

Observe that

$$reg_L(S) = \begin{bmatrix} S_1 \\ S_2 \cdot L \end{bmatrix} = \begin{bmatrix} S_1 & 0 \\ 0 & S_2 \end{bmatrix} \cdot \begin{bmatrix} I \\ L \end{bmatrix}$$

therefore

$$reg_L(S') \cdot (reg_L(S))^\dagger = \tag{21}$$

$$= \begin{bmatrix} S'_1 & 0 \\ 0 & S'_2 \end{bmatrix} \cdot \begin{bmatrix} I \\ L \end{bmatrix} \cdot \left( \begin{bmatrix} S_1 & 0 \\ 0 & S_2 \end{bmatrix} \cdot \begin{bmatrix} I \\ L \end{bmatrix} \right)^\dagger$$

$$= \begin{bmatrix} S'_1 & 0 \\ 0 & S'_2 \end{bmatrix} \cdot \begin{bmatrix} I \\ L \end{bmatrix} \cdot \begin{bmatrix} I \\ L \end{bmatrix}^\dagger \cdot \begin{bmatrix} S_1 & 0 \\ 0 & S_2 \end{bmatrix}^{-1}$$

$$= \begin{bmatrix} S'_1 & 0 \\ 0 & S'_2 \end{bmatrix} \cdot \begin{bmatrix} I \\ L \end{bmatrix} \cdot \begin{bmatrix} I \\ L \end{bmatrix}^\dagger \cdot \begin{bmatrix} S_1^{-1} & 0 \\ 0 & S_2^{-1} \end{bmatrix}$$

Since $\begin{bmatrix} I \\ L \end{bmatrix}$ is full rank, we have

$$\begin{bmatrix} I \\ L \end{bmatrix}^\dagger = \left( \begin{bmatrix} I \\ L \end{bmatrix}^T \cdot \begin{bmatrix} I \\ L \end{bmatrix} \right)^{-1} \cdot \begin{bmatrix} I \\ L \end{bmatrix}^T =$$

$$\begin{bmatrix} (I+L^2)^{-1} & | & L \cdot (I+L^2)^{-1} \end{bmatrix}$$

so we see that this implies equation 22. From equation 22 we see that the set of eigenvalues of $reg_L(S') \cdot (reg_L(S))^\dagger$ is not independent of $L$; indeed, if that were the case then the trace of the above matrix (i.e. the sum of its eigenvalues) would be the same for all $L$. However,

$$trace(reg_L(S') \cdot (reg_L(S))^\dagger) = \tag{23}$$

$$= \sum_i \frac{\text{diag}(S'_1 S_1^{-1})_{ii}}{1 + L_{ii}^2} + \sum_i \frac{L_{ii}^2 \text{diag}(S'_2 S_2^{-1})_{ii}}{1 + L_{ii}^2}$$

which is constant if and only if $\text{diag}(S'_1 S_1^{-1}) = diag(S'_2 S_2^{-1})$ (just take partial derivatives with respect to $L_{ii}$).

$\square$

**Claim 2**. *The matrix $\mathcal{K}_r$ in equation 6 is in general, **not invariant** under $reg_L$.*

*Proof.* Let us choose $S, S'$ satisfying the assumptions of the proof of Claim 1. Let us also consider the case where $r$ is the full rank of $S$, so that $\mathcal{K}_r = \mathcal{K} = S'S^\dagger$. Let $(U, \Sigma, V)$ be a singular value decomposition of $S$. Further let $(U_L, \Sigma_L, V_L)$ be a singular value decomposition of $reg_L(S)$, that is the transformation of $S$ under

regional color changes operation $reg_L$ defined above. If equation $6$ were invariant under $reg_L$, then we would have -for all diagonal $L$-

$$U^* S' S^\dagger U = U_L^* \cdot reg_L(S') \cdot (reg_L(S))^\dagger \cdot U_L$$

which would imply that

$$trace(U^* S' S^\dagger U) = trace(U_L^* \cdot reg_L(S') \cdot (reg_L(S))^\dagger \cdot U_L)$$

and therefore

$$trace(S' S^\dagger) = trace(reg_L(S') \cdot (reg_L(S))^\dagger)$$

for all diagonal $L$, since $trace(AB) = trace(BA)$ for any matrices $A, B$ and since $U^* U = U_L^* U_L = I$.

However as we saw in the proof of Claim 1, $trace(reg_L(S') \cdot (reg_L(S))^\dagger)$ is not independent of $L$; a contradiction.
□

