# OpenReview forum: "Self-supervised Color Generalization in Reinforcement Learning"
_TMLR — Accepted by TMLR_

### Review · Reviewer_PWdg · 2024-08-01

**Summary Of Contributions:**

This paper introduces a self-supervised approach to improve color generalization in reinforcement learning. Specifically, a Color-invariant Layer (CiL) has been introduced, which incorporates spatial and temporal permutation invariance. Compared to previous data augmentation methods such as random-conv and color-jitter, the proposed approach shows better performance in achieving color generalization while allowing color sensitivity.

**Audience:**

Yes

**Claims And Evidence:**

Yes

**Requested Changes:**

Please refer to the section above.

**Strengths And Weaknesses:**

### Strengths
* The research question studied in this paper is of importance to the community. Preserving color sensitivity is critical to achieve better generalization in many real-world tasks (e.g. autonomous driving).

### Weaknesses and other questions
* I don't quite understand how the proposed method can preserve color sensitivity (specifically Eqn. (7)). I would appreciate the authors could provide some intuitive explanations.
* From Figure 3, it seems that the benefits could also come from patch-wise color augmentation. Have the authors done such ablation experiments?
* Figure 4: how are "useful" and "less-useful" defined?
* Figure 1 (right-bottom subfigure): how to interpret the two axes?
* For the MiniGrid experiment in Section 4.1, is training and testing on the same environment?

---

> ### Author Response · Authors · 2024-09-09
> **Response to Reviewer PWdg**
>
> We appreciate the reviewer's valuable feedback and suggestions.
>
> ---
>
> **I don't quite understand how the proposed method can preserve color sensitivity (specifically Eqn. (7)). I would appreciate the authors could provide some intuitive explanations.**
>
> This may be best illustrated with the simplest example. Let’s assume only a single feature ( with 3 colors) and  the weight matrix in eq.7 consists of weights such that the preserved symmetry is of category III. Then  only three distinct color transformations are equivalent, i.e if one would present those three color augmented images to the Cil layer it would map it to the same latent representation for the CNN. If it chooses one of CiL transformation to be close to the identity that means that the CNN receives  the same comparable  information as without CiL present. Thus it can be color sensitive but also generalize to the two other color augmentations. However, we have no formal proof that such an color-sensitive representation always exist. This is an empirical result.
>
> ---
>
> **From Figure 3, it seems that the benefits could also come from patch-wise color augmentation. Have the authors done such ablation experiments?**
>
> There seems to be a misunderstanding of Figure 3, the different colors of patches are for illustration purposes only. Those patches are not independent as the same color invariance is learned for each feature across all patches, i.e. the weights of CiL are shared for each patch. We have added a clarification to the rebuttal.
>
> The ablation study proposed by the reviewer is interesting, and we have added a paragraph to discuss it further: *”One might wonder if patch-wise color augmentation could yield similar results to CiL. However, empirical tests on Minigrid show that patch-wise augmentations, like global color augmentations, also fail to learn effectively. Additionally, to achieve the same invariance properties as CiL, the model would require an impractical number of patch-wise augmentations, namely: #local color augmentations^#patches.”*
>
> ---
>
> **Figure 4: how are "useful" and "less-useful" defined?**
>
> We have rephrased the description : ”Comparison of symmetries which aid learning (a) and (b) contrasted to those  impeding the ability to learn crucial  features (c) and (d)”.
>
> ---
>
> **Figure 1 (right-bottom subfigure): how to interpret the two axes?**
>
> We have extended the description of the Figure 1: *“...OUR self-supervised learning of color-symmetries (right), in the example of Category II color symmetries -horizontal axis-  per each feature of the model - vertical axis-.   The vertical axis shows different features-wise choices of Category II symmetries. The horizontal axis shows identical images under the particular Category II symmetry. In other words, CiL produces the same feature-wise output if the input image varies according to horizontal axis.“*
>
> ---
>
> **For the MiniGrid experiment in Section 4.1, is training and testing on the same environment**
>
> First, the primary purpose of Figure 5b is to demonstrate that, in color-sensitive environments, image augmentation causes the agent to fail in learning, whereas our method performs comparably to the baseline. Second, the only color-generalization test conducted on Minigrid is shown in Figure 7b.

---

> > ### Comment · Reviewer_PWdg · 2024-09-14
> > **Thank you**
> >
> > Thank the authors for clarifying and addressing my questions.

---

### Review · Reviewer_P3Da · 2024-08-17

**Summary Of Contributions:**

This paper is about generalization to color variations when learning a reinforcement learning policy. In particular the authors use learned color invariant features by using the singular value decomposition and adding some trainable weights to account for different kinds of invariances, which can be configured by the user.

Results show that in a challenging setting (modified lavacrossing) the proposed method learns better policies that obtain higher accumulated rewards, and this is also confirmed in the procgen and distracted deepmind control benchmarks. Additional experiments in modified lavacrossing confirm that the proposed method learns different kinds of color invariances that are useful in a reinforcement learning setting.

Contributions and new knowledge are:
- The CiL layer for color invariant feature learning in reinforcement learning.
- A conceptual framework for learnable color invariant features that considers three kind of invariances, that can be set by the user.
- Experimental results showing that the CiL layer improves performance by learning color invariant features in a three environments using DQN and PPO.

**Audience:**

Yes

**Broader Impact Concerns:**

The paper presents a generic impact statement, and I think its insufficient to claim no impact. Feature learning definitely has an impact, as the invariances to color can have negative societal impact, just think about skin color, ideally one wants a model to ignore skin color, but depending on the task skin color could be required, and being invariant to it could reduce performance, so definitely there can be ethical implications of this work that could be discussed, as invariances are not neutral. Another example is to simply learn the wrong invariance that works in train/val/test sets, but completely fail in an open set setting  in the real world.

**Claims And Evidence:**

Yes

**Requested Changes:**

- Please clarify how this method is self-supervised. To me this is not clear from the paper, i.e. what exact formulation or mechanism makes the model self-supervised, and how did you validate this?
- I note that experiments show that categories I, II, and III, provide different reward performance, but something that is not clear from the practical/implementation point of view, how should practitioners select which category to use? Should it just be
- For validation purposes, can you provide samples of the learned invariant features? I would expect different inputs with color variations, and the learned invariant features that are output by CiL in different settings.
- How is patch size chosen? Of course this would be task specific, and at least the paper should mention how the patch sizes were chosen, so practitioners can tune the patch size for their specific task. I suggest this could be described in the appendix.
- It would be nice to have clear claims/contributions statements in the papers' introduction, and an audience statement.

**Strengths And Weaknesses:**

Strengths
- The paper is well written and understandable and presented.
- The paper is very well connected to the literature and references all relevant papers.
- The problem is important, as some degree of color invariance is required in all learned policies, depending on the environment and task, which should improve robustness.
- The mathematical formulation and explanation of the proposed method is solid, I only have minor remarks for improvement here.
- The baselines are color jitter and random conv, seems appropriate for the problem of color invariance, and the proposed method outperforms the baselines with ease.

Weaknesses
- While I understand the formulation for color feature invariance, to me it is not clear why the method is claimed as self-supervised, there is no description of the self-supervised part in the paper, its not clear to me how this method is self-supervised in the common meaning in ML and CV, compared to autoencoders and contrastive learning for example. While the invariance is definitely learned according to the task/environment, maybe the method cannot be called self-supervised.
- The results are limited as they show the proposed method obtaining better reward, but there is no direct validation about color invariance in the more complex settings (procgen and distracted dm control). Here it would have been nice to provide some visualizations to show the color invariance in multiple settings.

---

> ### Author Response · Authors · 2024-09-09
> **Response to Reviewer P3Da**
>
> We appreciate the reviewer's valuable feedback and suggestions.
>
> ---
>
> **Please clarify how this method is self-supervised…**
>
> We acknowledge that 'self-supervised' is used somewhat differently here. In conventional training, an agent is trained with randomly color-augmented images (see Figure 1), and task-specific subsets can be selected to improve performance. Our method automates the selection of 'useful' color augmentations in a self-supervised manner, making the term appropriate. Additionally, CiL introduces different inductive biases, with the most suitable bias being selected during training in a self-supervised fashion.
>
> ---
>
> **I note that experiments show that categories I, II, and III, provide different reward performance, but something that is not clear from the
> practical/implementation point of view, how should practitioners select which category to use?**
>
> We have added the following paragraph:  *“The primary conclusion of the paper is that a mixture of different categories yields the best performance. The general principle is that by incorporating all categories—each with distinct inductive biases—the network is better equipped to select the most appropriate features.”*
>
> ---
>
> **For validation purposes, can you provide samples of the learned invariant features?**
>
> This is a great  suggestion. We had originally attempted such a sample but the resulting invariances were not human interpretable and looked similar to Figure 2, thus we chose to omit it. In a way this is not surprising as the output of CiL has many features; the agent model learns by contrasting different features, i.e. it contrasts different symmetry invariant representations.
> The validation of the method is given by the formal proofs. These show that the inductive biases/ symmetries are indeed learned by the method. That those in general represent color augmentation is shown in Figure 1 and 2 where we do illustrate those transformations.
>
> ---
>
> **How is patch size chosen? Of course this would be task specific, and at least the paper should mention how the patch sizes were chosen, so practitioners can tune the patch size for their specific task. I suggest this could be described in the appendix.**
>
> We have added a brief discussion in the appendix . *"The patch size follows the convention in Vision Transformers; for a 64x64 pixel input, we use 8x8 patches. In Figure 3, the neighborhood sizes are chosen as odd numbers (3, 5) to ensure a central patch with an equal number of neighboring patches on all sides."*
>
> ---
>
> **It would be nice to have clear claims/contributions statements in the papers' introduction, and an audience statement.**
>
> We have extended our contribution statements in the introduction to highlight why the method is self-supervised, and mention the theoretical and empirical results.
>
> ---
>
> **The paper presents a generic impact statement, and I think its insufficient to claim no impact.**
>
> We thank the reviewer  for their suggestion.  We have extended our impact statement. *”...In particular, while improved color-sensitivity while maintaining color-generalisation is  desirable for many tasks , there are risks, such as an agent learning to make decisions based on an individual's skin color. However, we conclude that such a behavior would likely arise from biases in the training dataset or environment, rather than from our method itself.”*

---

### Review · Reviewer_4MkC · 2024-08-28

**Summary Of Contributions:**

This paper presents a novel neural network based on the Dynamic-mode-decomposition algorithm to improve color generalization in image-based reinforcement learning. Three different categories of color symmetries are introduced in the paper. Each color-symmetry category can be achieved by determining the learnable parameters matrix of the neural network. The paper provides insightful discussions about the unuseful color symmetry. The authors perform empirical evaluations on the modified Lavacrossing environment, and the environments from Procgen, and DeepMind Control suites.

**Audience:**

Yes

**Broader Impact Concerns:**

No concerns on the ethical implications.

**Claims And Evidence:**

Yes

**Requested Changes:**

- Providing additional statistically significant results to support the statement in Subsection 4.1 that CiT improved the performance than the CNN on the modified lava-crossing environment.

- Additional experiments are suggested to compare CiT with the random shift image augmentation by DrQ-v2.

- Presenting error bars for the Procgen results shown in Fig.9, Table 1 and Table 2.

- Explaining the meaning of the error bars in Fig.5 (b).

- It would help to reproduce and understand the work by providing more implementation details, such as those of the baselines and the RL algorithms mentioned in subsections 4.1 and 4.2.

[1] Yarats, Denis, et al. "Mastering visual continuous control: Improved data-augmented reinforcement learning." arXiv preprint arXiv:2107.09645 (2021).

**Strengths And Weaknesses:**

**Strengths**:
- The paper is well-motivated and well-organized.

- The authors provide a theoretical proof demonstrating the symmetry invariance of the proposed method.

- The classification of color symmetries and their corresponding visualizations (in Fig.2) are interesting.

- The authors conducted empirical evaluations across several benchmarks.

**Weaknesses:**

- The main concern is that the experimental results do not demonstrate the superiority of the proposed method over the baselines: 1) Regarding the results on the modified Lavacrossing environment (see Fig.5(b)), the shading area of the CNN baseline overlaps with the shading of category II or category III.  2) In Table 3, the scores achieved by CiL are only comparable to the CNN baseline on the two DMC tasks. 3) Furthermore, the authors did not report error bars of the scores obtained on the Procgen tasks (see Fig.9, Table 1 and Table 2). Therefore, it is insufficient to draw the conclusion that CiL enhanced performance and sample efficiency than baselines.

-  The random shift image augmentation presented in DrQ-v2 [1] has achieved promising results in image-based RL tasks. Comparing CiL to DrQ-v2 could improve the paper’s impact. However, experiments comparing CiL to it are missing.

- The three distinct categories perform differently across tasks. Discussions about how to choose the CiL category for the specific task in practice are suggested to be added.

- The authors show the results of CiL with two categories (in Figure 7). I recommend the authors to add implementation details about how to use two categories at a time and discussion about when to use CiL with two categories rather than one category.

---

> ### Author Response · Authors · 2024-09-09
> **Response to Reviewer 4MkC**
>
> We appreciate the reviewer's valuable feedback and suggestions.
>
>
> ---
>
> **The main concern is that the experimental results do not demonstrate the superiority of the proposed method over the baselines…**
>
> For the main results on Procgen, the standard error is provided in the description of the respective table. We've highlighted this in the rebuttal to ensure it is not overlooked in the future: *"The standard error for CNN+CiL as well as the baseline is <0.2% for each game individually."* Moreover, we used the evaluation method from Agarwal et al., which reduces statistical fluctuations, ensuring that the performance gains are statistically significant.
>
>
> We would like to address some misunderstandings. First, the primary purpose of Figure 5b is to demonstrate that, in color-sensitive environments, image augmentation causes the agent to fail in learning, whereas our method performs comparably to the baseline. Second, the only color-generalization test conducted on Minigrid is shown in Figure 7b. The results are statistically significant, as the standard error bars in Figure 7b for the generalization test do not overlap.
>
> ---
>
> **The three distinct categories perform differently across tasks. Discussions about how to choose the CiL category for the specific task in practice are suggested to be added.**
>
> We have added the following paragraph:  *“The primary conclusion of the paper is that a mixture of different categories yields the best performance. The general principle is that by incorporating all categories—each with distinct inductive biases—the network is better equipped to select the most appropriate features.”*
>
> More detailed discussions can be found in the final paragraphs of sections 4.1 and 4.2.
>
> ---
>
>
> **The random shift image augmentation presented in DrQ-v2 [1] has achieved promising results in image-based RL tasks. Comparing CiL to DrQ-v2 could improve the paper’s impact..**
>
> The suggested paper focuses on continuous control, and we have included it as a reference. However, our baseline, DrAC + crop, is similar in nature, and we are unclear on why the DrQ-v2 baseline is crucial for our work. Could the reviewer please elaborate on why it is important to demonstrate the significance of our results using this baseline?
>
> ---
>
> **...I recommend the authors to add implementation details about how to use two categories at a time...**
>
> The implementation of using different categories is straightforward; it is achieved by concatenating features with symmetry properties from categories I, II, and III, respectively.

---

> ### Comment · Reviewer_4MkC · 2024-09-09
> **Official Comments by reviewer 4Mkc**
>
> >  The standard error for CNN+CiL as well as the baseline is <0.2%
>
> I thank the authors for pointing out my oversight. For the provided results on the Lavacrossing environment (see Fig. 5b) and 2 DMC tasks (Table 3), as well as 3 among 5 procgen tasks (Table 1) , CiL is only comparable to the baselines. So I still think that the current experimental results are too weak to prove the benefits of CiL.
>
> > <Results in Figure 5b>
>
> I would like to clarify my concerns about Figure 5b. Regarding the results in Fig.5b, there are some inaccurate statements in section 4.1, such as "when we incorporated an initial CiL with the CNN for color-symmetry categories I, II, and III, we observed improved performance."  In my view, CNN without data augmentations achieves comparable performance than CNN with category II and III, while being better than CNN with category I.
>
> > why it is important to demonstrate the significance of our results using this baseline?
>
> Firstly, I don't think the crop mentioned in the paper does the same thing as the data augmentation proposed by Drq-v2.
> The image augmentation presented in Drq-2 is implemented by firstly padding images by 4 pixels and then randomly cropping the padded image to the original size. A bilinear interpolation technique is also added to the data augmentation to improve performence.
>
> Secondly, the data agumentations presented in Drq-v2 has been proved to greatly  improve the performance on image-based RL tasks. In my opinion, Drq-v2 or Drq is a milestone in data-augmentation for RL. This paper does the same thing with Drq-v2 but lacks a comparison to it.
>
> So, I thought that adding comparison to the Drq-v2 data augmentation is neccessary and showing benefits over it would increase the impact of this work.

---

> ### Author Response · Authors · 2024-09-10
> **Add-On**
>
> We appreciate the reviewer's swift response.
>
> ---
>
> **Regarding the results in Fig.5b, there are some inaccurate statements in section 4.1,**
>
> We have modified this single sentence of concern to: *"In contrast, when we incorporated CiL with the CNN for color-symmetry categories I, II, and III, learning proceeded without obstruction."*
>
> We would like to emphasize  that  already in the introduction we state: *"In section 4.1, we show that in the simple example of adding a safe blue river (water) to the mini-grid LavaCrossing environment data augmentation - random-conv and color-jitter -  is detrimental to learning, while  CiL  is able to navigate the color-sensitive environment."*
>
> ---
>
> **The image augmentation presented in Drq-2 is implemented by firstly padding images by 4 pixels and then randomly cropping the padded image to the original size.**
>
> That is exactly what crop means for DrAC as well, i.e. padding by 12x12 pixels and then crop of 64x64 to original image-size.  (As for the scaling with a bilinear transformation that is not considered in our approach.)
>
> We do agree that if our focus would be on DMControl evaluation a Drq-2 baseline would be desirable. However, our goal is different. We provide  a proof-of-concept for CiL's ability to generalize to background videos in DMControl. This successful test is significant to us, as SVD/DMD can be highly sensitive to dynamic changes introduced by background videos. The DMControl test demonstrates CiL's effective generalization to new data distributions relevant for real-world tasks, rather than aiming to outperform state-of-the-art (SOTA) methods. We have added a paragraph to the work to emphasize this point.
>
> ---
>
> **the current experimental results are too weak to prove the benefits of CiL.**
>
> The paper claims that CiL offers generalization benefits in environments where color sensitivity is crucial. We believe our results support these claims—does the reviewer agree? This specifically refers to the main generalization tests on Minigrid (Figure 7) and the diverse Procgen generalization tasks, particularly in the three tasks where color-data augmentation fails.

---

### Decision · Action_Editor_DvVv · 2024-10-08

**Recommendation:** Accept with minor revision

**Comment:**

Reviewer 4MkC and PWdg were concerned about the practical benefits, since the practical benefits compared to existing method for RL from images that do not explicitly address color invariances have not been demonstrated. Reviewer PWdg is leaning to reject for this reason. However, as all reviewers agree that the claims are substantiated and the findings relevant for some of TMLR's audience, I argue that the paper can be accept.

I request a very minor revision: The citations in the paper are not properly formatted. For the camera-ready submission, please use \citep{} to put the full citation in parentheses, unless the citation is used as a noun in the sentence.

**Audience:**

The paper proposes a novel method to learn task-specific color invariances for image-based reinforcement learning, and demonstrate that this approach can help learning better policies on tasks where color invariances are important. These finding seem interesting to some of the audience.

**Claims And Evidence:**

The main claim of the paper regarding the color invariances of the proposed layers are supported by proofs and empircal evaluations. All reviewers state that the claims are sufficiently well supported.